# The persistence potential of transferable plasmids

Teng Wang[1] & Lingchong You [ORCID] [1,2,3 ✉]

Conjugative plasmids can mediate the spread and maintenance of diverse traits and functions in microbial communities. This role depends on the plasmid's ability to persist in a population. However, for a community consisting of multiple populations transferring multiple plasmids, the conditions underlying plasmid persistence are poorly understood. Here, we describe a plasmid-centric framework that makes it computationally feasible to analyze gene flow in complex communities. Using this framework, we derive the 'persistence potential': a general, heuristic metric that predicts the persistence and abundance of any plasmids. We validate the metric with engineered microbial consortia transferring mobilizable plasmids and with quantitative data available in the literature. We believe that our framework and the resulting metric will facilitate a quantitative understanding of natural microbial communities and the engineering of microbial consortia.

[1] Department of Biomedical Engineering, Duke University, Durham, NC 27708, USA. [2] Center for Genomic and Computational Biology, Duke University, Durham, NC 27708, USA. [3] Department of Molecular Genetics and Microbiology, Duke University School of Medicine, Durham, NC 27708, USA. ✉email: you@duke.edu

Mobile genetic elements (MGEs), including plasmids, transposons and phages are major components of the metagenome of microbes[1,2]. Hundreds of plasmids have been identified in diverse microbial communities in different environments such as rat cecum[3], cow rumen[4], sludge[5], and marine water[6]. In the reference genomes of the human gut microbiome, ~16,000 genes were identified as mobile[7]. MGE-associated genes encode diverse biological functions like metabolic capabilities[8], pathogenic virulence[9], plasmid addiction[10], or traits to cope with environmental stresses[11–13]. The interplay between MGEs and the core genomes of the host cells shapes the evolution of microbial communities[14].

The ability for a transferable plasmid (by conjugation) to persist in a microbial community can influence the dynamics, function, and even survival of the community[15,16]. Promotion or suppression of plasmid persistence, depending on the context, has applications in medicine, biosafety, and biotechnology. For instance, eliminating a plasmid that encodes the resistance to an antibiotic can inhibit the spread of the plasmid-associated resistance genes[17], which can enable more effective use of the antibiotic[18]. There is always a biosafety concern about the risk associated with the spread of synthetic genetic constructs into the environment[19,20]. To reduce the risk, the use of plasmid vectors with restricted capability to be maintained outside of the laboratory has been proposed[19,20]. In biotechnology, plasmid instability is a major impediment to the large-scale production of recombinant protein products[21,22]. The expression of recombinant proteins encoded by the plasmids are usually burdensome to the cell metabolism, and, as a consequence, the plasmid-carrying cells can be outcompeted by the faster-growing plasmid-free populations. A strategy to promote plasmid persistence could overcome this limitation.

The quantitative studies of plasmid persistence and abundance in microbial communities is challenging due to the lack of an effective computational framework[23,24]. Since the 1970s, population-biology models have been developed to predict the persistence of a single plasmid in a single species[17,25–27]. However, microbes in nature often live in complex communities consisting of diverse species and plasmids[7,28,29]. The limited scope of past modeling or experimental analyses is due in part to the computational challenge associated with modeling complex communities using the conventional modeling framework, which we refer to as 'subpopulation-centric framework' (SCF). In SCF, a population carrying a particular combination of plasmids is considered a unique subpopulation that requires one ordinary differential equation (ODE) to describe. Modeling a community containing two species and two plasmids requires eight ODEs, each describing one subpopulation. The model complexity increases combinatorially with the number of plasmids (Fig. 1a, b). For example, the marine microbiome in a bottle of sea water is estimated to contain ~160 species[29] and ~180 plasmids[6]. In SCF, ~$2.5 \times 10^{56}$ ODEs and ~$2.7 \times 10^{114}$ parameters are needed to model the gene flow dynamics of this community. A model with such complexity would far exceed the current combined computational power of the entire world[30].

In this work, we develop a plasmid-centric framework (PCF) to overcome the computational challenge associated with the SCF. Compared with SCF, PCF drastically reduces the model complexity and makes it computationally feasible to analyze plasmid transfer dynamics in complex communities. Using PCF, we derive a metric, termed persistence potential, to predict the plasmid persistence and abundance. Our experiments with engineered microbial communities transferring mobilizable plasmids and data in literature demonstrate the general predictive power of the persistence potential.

## Results

**A plasmid-centric framework to model gene flow.** To overcome the challenge associated with SCF, we developed a plasmid-centric framework (PCF) that focuses on the overall abundance of each plasmid in the community by accounting for the average growth effect of plasmids to the host. To illustrate the key concepts, consider $n$ types of transferable plasmids in a community consisting of $m$ species. Here, the transferable plasmids are the ones that can be transferred by conjugation. Let $s_i$ ($i = 1, 2, …, m$) be the abundance of $i$-th species and $p_{ij}$ ($j = 1, 2, …, n$) the abundance of the $i$-th species carrying the $j$-th plasmid (Supplementary Fig. 1a). The abundance was defined as the cell density of each population. The community dynamics can be approximately described by two groups of ODEs

$$\frac{ds_i}{dt} = \alpha_i \mu_i^e s_i - D s_i \qquad (1)$$

$$\frac{dp_{ij}}{dt} = \beta_{ij} \mu_{ij}^e p_{ij} + (s_i - p_{ij}) \sum_{k=1}^{m} \eta_{jki} p_{kj} - (\kappa_{ij} + D) p_{ij}. \qquad (2)$$

Equation (1) describes the collective growth and dilution of $s_i$, where $\mu_i^e$ is the effective growth rate of the species. Here we assumed each species follows logistic growth, in competition with other species sharing the same niche. Thus, $\mu_i^e$ accounts for the maximum growth rate and the carrying capacity: $\mu_i^e = \mu_i(e_i - \sum s_k)$, where $\mu_i$ is the maximum growth rate, $e_i$ is the carrying capacity, and $s_k$ is the density of the $k$-th species that shares the same niche. In light of the previous works[17,26], we considered a microbial community as an open system where flows enter the community and leave the community with resources, waste and cells. The dilution rate $D$ is the rate of the flow through the habitat as measured in turnovers per hour. $\alpha_i$ represents the average growth effect of all the plasmids carried by the species.

In Eq. (2), the first term describes the growth of $p_{ij}$. $\mu_{ij}^e$ is the effective growth rate; it differs from $\mu_i^e$ due to the growth effect of the $j$-th plasmid. $\beta_{ij}$ is the average growth effect of the other plasmids. In the second term, $\eta_{jki}$ is the conjugation efficiency when the plasmid is transferred from the $k$-th species to the $i$-th species (Supplementary Fig. 1a). $s_i - p_{ij}$ is the total abundance of subpopulations of $s_i$ not carrying the $j$-th plasmid. The third term describes the plasmid loss due to segregation error (at a rate constant of $\kappa_{ij}$) and dilution.

The average growth effect of the plasmids was calculated from the individual growth effect of each plasmid. Let $\lambda_{ij}$ represent the individual burden of the $j$-th plasmid in the $i$-th species. $\mu_{ij}^e$ is linked with $\mu_i^e$ via $\mu_{ij}^e = \mu_i^e/(1 + \lambda_{ij})$. The plasmid is burdensome with positive $\lambda_{ij}$ and beneficial with negative $\lambda_{ij}$. Here, we considered the growth effect of the plasmid as being inversely proportional to its burden. Therefore, the average growth effects were determined as $\alpha_i =$

$\frac{s_i}{s_i + \sum_{j=1}^{n}(p_{ij}\lambda_{ij})}$ and $\beta_{ij} = \frac{s_i(1+\lambda_{ij})}{s_i(1+\lambda_{ij}) + \sum_{(k:k \neq j)}(p_{ik}\lambda_{ik})}$.

$p_{ij}/s_i$ indicates the relative abundance of the $j$-th plasmid in the $i$-th species; $\frac{\sum_i p_{ij}}{\sum_i s_i}$ is the relative abundance of the $j$-th plasmid in the entire community. Because of the formulations of $\alpha_i$, $\beta_{ij}$, $\mu_i^e$ and $\mu_{ij}^e$, this framework is applicable for describing arbitrary microbial communities transferring multiple plasmids. In particular, the interactions between populations can be accounted for by the appropriate formulation of $\mu_i^e$ and $\mu_{ij}^e$; plasmid incompatibility can be accounted for by adapting the formulations of $\beta_{ij}$

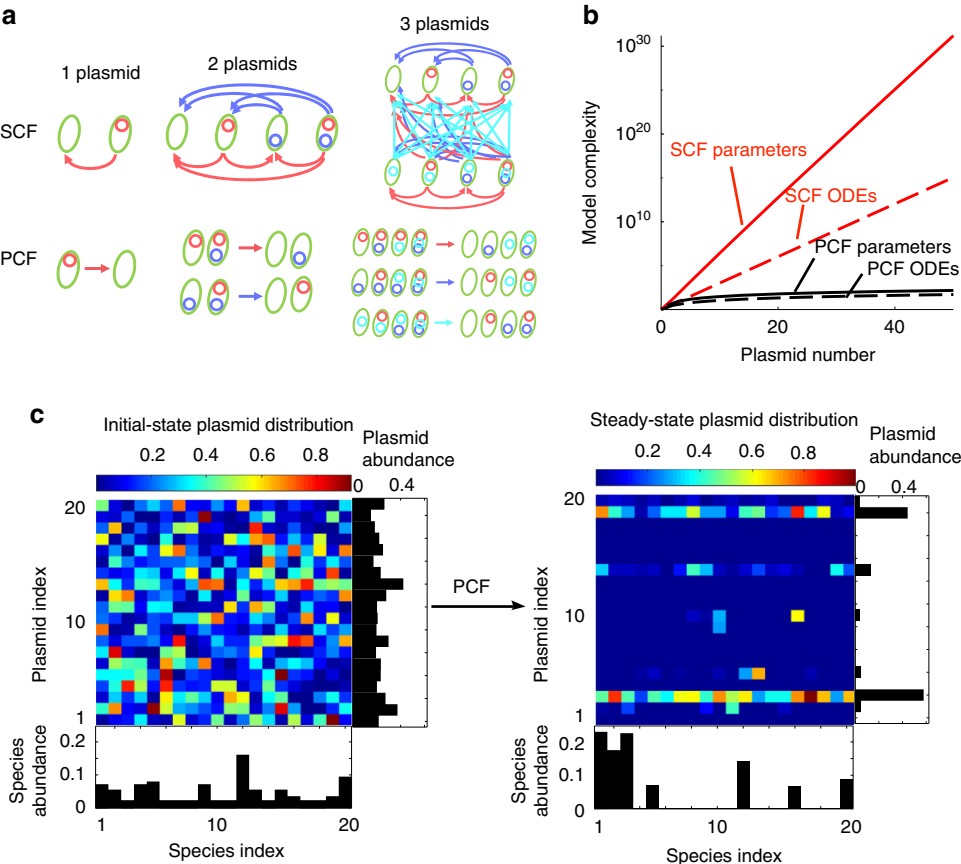

**Fig. 1 A plasmid-centric framework (PCF) to model horizontal gene transfer in microbial communities. a** Comparison between the subpopulation-centric framework (SCF) and PCF for one species transferring 1–3 plasmids. The arrows represent the plasmid transfer from the donor to the recipient. **b** Model complexity of single-species communities as a function of the number of plasmids. The model complexity refers to the number of ordinary differential equations (ODEs) or the number of parameters required in each model. **c** Simulation of the dynamics of a community consisting of 20 species transferring 20 plasmids. Different species or plasmids are distinguished by the indices. A random set of parameter values were used. The initial distributions (left) of population sizes and plasmid abundances were also random. The right panel shows the steady-state distributions of the plasmids across different species, quantified as the relative abundance of plasmids in each species. The relative abundance of plasmid $j$ in species $i$ was calculated as the fraction of species $i$ cells that contains plasmid $j$ relative to the total number of species $i$ cells. The total abundances of each species and each plasmid in the entire community are shown as bars.

and the conjugation terms (Supplementary Fig. 1b, see Supplementary Information section 2.1.2).

For a community consisting of $m$ species transferring $n$ plasmids, the SCF requires $m \cdot 2^n$ ODEs and approximately $nm^2 \cdot 2^{2n-2}$ parameters, whereas our PCF only needs $m(n+1)$ ODEs and approximately $nm^2$ parameters (Fig. 1a, b, Supplementary Table 2, see Supplementary Information section 2.2). This simplification enables our framework to compute the dynamics of species composition, as well as the distribution patterns of each plasmid (Fig. 1c). In particular, we modeled a community consisting of 200 species transferring 200 plasmids. The conventional SCF requires about $3.2 \times 10^{62}$ ODEs and $5.2 \times 10^{126}$ parameters, which is impractical to construct. In contrast, this community can be modeled by about $4.0 \times 10^4$ ODEs and $8.1 \times 10^6$ parameters using PCF, which is feasible to both construct and calculate (Supplementary Fig. 1c). Although acquiring all the parameters in such complex communities remains challenging due to current technical limitations[31], PCF enables the theoretical and computational analysis of the dynamics and persistence conditions of plasmids.

This drastic reduction in the model complexity is made possible by combining multiple distinct subpopulations into an average one and then accounting for the average growth effect of each plasmid. In particular, $s_i$ includes all subpopulations carrying

different combinations of plasmids or no plasmids and $p_{ij}$ includes all subpopulations carrying the $j$-th plasmid. If the plasmids do not have any growth effects, the PCF is equivalent to SCF (Supplementary Fig. 2a). In general, however, plasmids can confer burden or benefit, which will cause deviation between these two frameworks. To evaluate this discrepancy, we conducted numerical simulations on communities transferring one and two plasmids. Testing higher number of plasmids in SCF is computationally prohibitive due to the combinatorial explosion. Indeed, simulation results suggest that the growth effects of the plasmids are the main factors that determine the discrepancy. Within a reasonably wide range of fitness costs, the predictions of these two models match well, and the smaller the growth effects, the smaller the discrepancy (Supplementary Fig. 2, see Supplementary Information section 2.3). In general, we note that both frameworks represent approximations of real biological systems. The ultimate test of each framework should be from experiments.

**The persistence potential of plasmids.** Our framework makes it feasible to develop a criterion of plasmid persistence for complex communities. We first considered an idealized community, where each species has the same set of kinetic parameters and equal abundance (Fig. 2a). Because of its symmetry, this community allows the analytical derivation of a metric ($\omega$) that determines

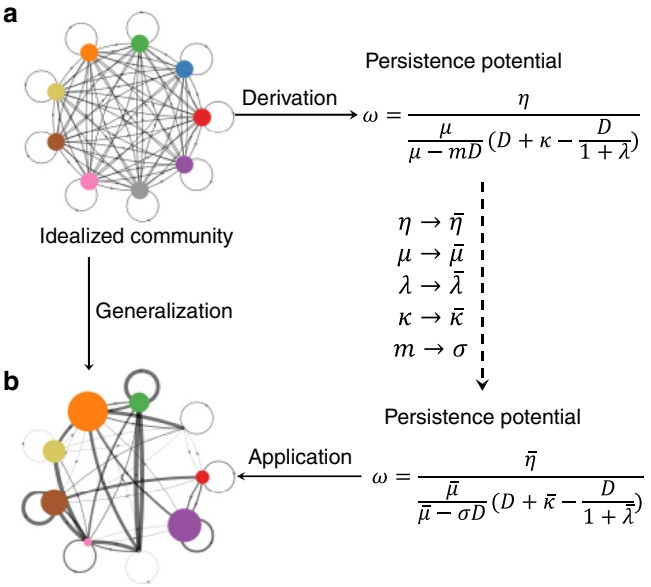

**a**

Idealized community

Generalization

**b**

Complex community

Persistence potential

Derivation

$$\omega = \frac{\eta}{\frac{\mu}{\mu - mD}\left(D + \kappa - \frac{D}{1+\lambda}\right)}$$

$$\eta \to \bar{\eta}$$
$$\mu \to \bar{\mu}$$
$$\lambda \to \bar{\lambda}$$
$$\kappa \to \bar{\kappa}$$
$$m \to \sigma$$

Persistence potential

Application

$$\omega = \frac{\bar{\eta}}{\frac{\bar{\mu}}{\bar{\mu} - \sigma D}\left(D + \bar{\kappa} - \frac{D}{1+\bar{\lambda}}\right)}$$

**Fig. 2 Developing the persistence potential ($\omega$) of plasmids. a** Derivation of $\omega$ for an idealized microbial community. Here, the community is fully symmetric in terms of parameters and population sizes. Each colored circle represents a constituent species; the area of the circle is proportional to the species size. Each directed arrow represents the transfer of a plasmid, with the arrow thickness indicating the transfer rate. In this case, $\omega$ can be analytically derived. **b** Generalization of $\omega$. We generalized the formulation of $\omega$ by replacing each parameter with its weighted average, which factors in the heterogeneity of species size. The weighted averages of $\mu$, $\kappa$, and $\lambda$ were calculated by $\bar{\mu} = \sum_{i=1}^{m} \frac{s_i}{s_T}\mu_i$, $\bar{\kappa} = \sum_{i=1}^{m} \frac{s_i}{s_T}\kappa_i$, $\bar{\lambda} = \sum_{i=1}^{m} \frac{s_i}{s_T}\lambda_i$, where $m$ is the total number of species and $s_i$ represents the abundance of $i$-th species. $s_T$ is the total abundance of all the species: $s_T = \sum_{i=1}^{m} s_i$. Both donor and recipient cell densities contribute to the conjugation efficiency; thus, the weighted average of $\eta$ was calculated by $\bar{\eta} = \sum_{i=1}^{m}\sum_{j=1}^{m} \frac{s_i}{s_T}\frac{s_j}{s_T}\eta_{ij}$, where $s_i$ represents the abundance of the donor species, $s_j$ the recipient species, and $\eta_{ij}$ the transfer rate from the donor to the recipient. In the formulation of the generalized persistence potential, the species number $m$ was replaced with the Shannon effective number of species, $\sigma$, which is calculated through $\sigma = e^{-\sum_{i=1}^{m}\frac{s_i}{s_T}\ln\frac{s_i}{s_T}}$.

the persistence and steady-state abundance of the plasmids relative to the total cell density (Fig. 2a, also see Supplementary Information section 2.4)

$$\omega = \frac{\eta}{\frac{\mu}{\mu - mD}\left(D + \kappa - \frac{D}{1+\lambda}\right)}. \qquad (3)$$

We termed this metric the persistence potential of the plasmid. Here, $\lambda$ represents the growth effect of the plasmid in the community. If $\omega < 0$, which means that the benefit of the plasmid overcomes the plasmid loss rate ($\lambda < -\frac{\kappa}{\kappa+D}$), the element will always persist. If the plasmid is burdensome ($\lambda > 0$), or the benefit of the plasmid alone is unable to overcome the plasmid loss ($-\frac{\kappa}{\kappa+D} < \lambda \le 0$), the value of $\omega$ will become positive. In this case, the plasmid can persist if and only if $\omega > 1$. With $0 < \omega < 1$, the plasmid will be lost. The interpretation of the persistence potential is similar to the basic reproduction number ($R_0$) in epidemiology: the disease will die out when $R_0 < 1$, and will become epidemic if $R_0 > 1$[32]. Another analogous example is the percolation threshold for the plague outbreak. The plague will become an outbreak only when the host abundance exceeds the percolation threshold[33].

In general, the constituent species in a community are not symmetric (Fig. 2b). Their abundances are different from each other; they have different growth rates; and they transfer plasmids at different rates. For such communities, deriving an analytical solution is not possible. We thus took a heuristic approach to generalize the idealized metric based on the intuition that species with greater abundance contributes more to the overall values of kinetic parameters. We thus kept the derived formulation but replaced each parameter with the weighted average of the corresponding parameter in the general community accounting for the relative abundances of different populations $\bar{\mu} = \sum_{i=1}^{m} \frac{s_i}{s_T}\mu_i$, $\bar{\kappa} = \sum_{i=1}^{m} \frac{s_i}{s_T}\kappa_i$, $\bar{\lambda} = \sum_{i=1}^{m} \frac{s_i}{s_T}\lambda_i$, and $\bar{\eta} = \sum_{i=1}^{m}\sum_{j=1}^{m} \frac{s_i}{s_T}\frac{s_j}{s_T}\eta_{ij}$, where $s_T$ is the total abundance of all the populations: $s_T = \sum_{i=1}^{m} s_i$. $\mu_i$ is the max growth rate of the $i$-th species. $\kappa_i$ and $\lambda_i$ are the loss rate and the burden of the plasmid in the $i$-th species, respectively. $\eta_{ij}$ is the transfer rate of the plasmid from the $i$-th to the $j$-th species. Due to the heterogeneity of the species abundances, the total number of species cannot describe the diversity of the community. For instance, if a community is dominated by one single species, while the other species only occupy extreme small abundances, the community will behave more like a single-species population instead of a multiple-species one. Therefore, we replaced the absolute number of species, $m$, with the Shannon effective number of species, $\sigma$, which was calculated through $\sigma = e^{-\sum_{i=1}^{m}\frac{s_i}{s_T}\ln\frac{s_i}{s_T}}$ [34]. The general form of persistence potential becomes:

$$\omega = \frac{\bar{\eta}}{\frac{\bar{\mu}}{\bar{\mu}-\sigma D}\left(D + \bar{\kappa} - \frac{D}{1+\bar{\lambda}}\right)} \qquad (4)$$

where $\bar{\mu}$, $\bar{\kappa}$, $\bar{\lambda}$, $\bar{\eta}$, and $D$ stand for the weighted averages of species growth rate, plasmid loss rate, fitness cost, horizontal transfer rate, and dilution rate, respectively (Fig. 3a, b). Based on this formulation, plasmid persistence can be promoted by increasing plasmid transfer rate or species growth rates, and be suppressed by increasing plasmid fitness cost, segregation loss or dilution rate (Fig. 3a). Because each component of plasmid persistence potential is the weighted average with respect to species abundance, modulating community composition can also change the persistence potential.

Here, we established the persistence potential by approximation. To evaluate the predictive power of the generalized formulation, we performed numerical simulations with 2000 sets of randomized parameters in communities consisting of 5–100 species and 1–50 plasmids. Our randomization process produced communities with various settings that mimic the diversity of the natural genetic-exchange communities. To ensure there were adequate numbers of species coexisting at steady states, each community was divided into a number of coexisting niches. A niche can arise from multiple populations consuming the same substrate[35]. Different substrates would lead to different niches. In our simulations, different populations in the same niche compete with each other by having a shared carrying capacity. In each simulation, the number of coexisting niches was randomized between 1 and the total number of species. Each species was randomly distributed into one of the niches and the carrying capacities of each niche was randomized. All the parameters followed uniform distributions in the given ranges (see Section 2.1.3 of the Supplementary Information for more details). When we performed the numerical simulations, we normalized the abundances with the combined capacity of all the niches. Thus, the abundance in our simulations was dimensionless. The dynamics of each community was simulated for up to ~30,000 h until it became unchanged, and we treated it as 'steady state'. The fractions of the plasmid-carrying cells were then

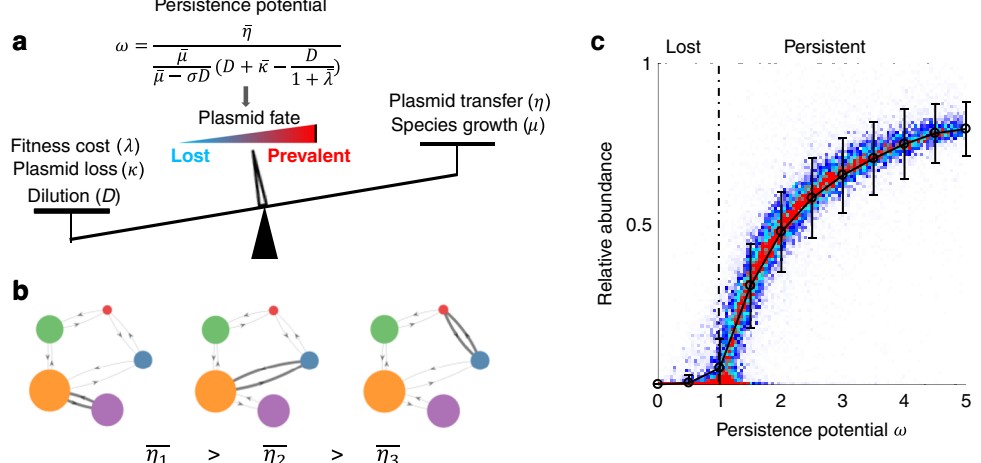

**Fig. 3 Persistence potential $\omega$ predicts the maintenance and abundance of plasmids. a** $\omega$ accounts for the contribution from multiple parameters. The plasmid horizontal transfer and population growth promote the $\omega$ value, while fitness cost, plasmid loss, and dilution suppress the $\omega$ value. **b** The weighted average transfer rates ($\bar{\eta}$) are calculated with respect to species abundances. A community of five species and one plasmid is illustrated as an example. The arrows represent the horizontal transfer of the plasmid, and the width of the arrow represents the magnitude of the transfer rate. The filled circles represent the populations, with their areas corresponding to the abundance. With equivalent conjugation rates, the conjugation pairs with higher donor or recipient abundance contribute more to $\bar{\eta}$. **c** $\omega$ is a predictor of the steady-state abundance of plasmids. 2000 simulations were performed with 5–100 species, 1–50 plasmids, and randomized parameters in the range of $0.4 \leq \mu \leq 0.8 \, h^{-1}$, $0.001 \leq D \leq 0.005 \, h^{-1}$, $0 \leq \kappa \leq 0.002 \, h^{-1}$, $0 \leq \lambda \leq 0.2$, $0 \leq \eta \leq 0.02 \, h^{-1}$. All randomized parameters followed uniform distributions in their ranges. For each simulation, the communities were assembled into a random number of niches. Within each niche, species compete with each other. Each simulation was initialized with random abundances of species and plasmids. The steady-state persistence potential $\omega$ of each plasmid and its relative abundance in the entire community were then calculated. The $\omega$ range was divided into multiple bins with widths of 0.5. The bar plots represent the mean values $+/-$ standard deviations of all the plasmid abundances within each bin. Each bin contains 1334 to 8547 independent replicates.

calculated with respect to $\omega$ values. Our simulation results suggest that the general metric, despite its heuristic nature, remains a predictor on whether and to what extent a plasmid can persist, with a transition at $\omega = 1$ (Fig. 3c). When $0 < \omega < 1$, the abundance of the plasmid is close to 0; when $\omega > 1$, the abundance of the plasmid increases monotonically with $\omega$. The data points computed from the randomized parameter sets approximately collapsed into a single curve, suggesting the general predictive power of $\omega$.

In addition to assuming symmetry, the basic form of $\omega$ (Eq. (4)) was derived by assuming that the system reached equilibrium state. However, this assumption is not critical for the approximate predictive power of $\omega$. In particular, in the numerical simulations, the $\omega$ values can have similar predictive power for the plasmid abundance well before the system has reached equilibrium state. One example was shown in Supplementary Fig. 3. This result underscores the general predictive power of $\omega$ and its applicability to experimental systems, which may not be at equilibrium state.

**Experimental validation of the persistence potential.** To test the predictive power of the persistence potential, we engineered eight communities transferring mobilizable plasmids. Communities 1 through 7 were constructed from three *E. coli* strains (denoted X, B, and R). Strain B expresses BFP on the chromosome, R expresses dTomato, and strain X is not fluorescent. The mobilizable plasmid K was transferred among the strains and expresses GFP constitutively. Using flow cytometry, this system allows the simultaneous quantification of plasmid abundance (with GFP) and population compositions (with BFP and dTomato) (Supplementary Fig. 4a). Community 8 contained two *E. coli* strains (MG1655 and DH5α) and five conjugative plasmids (F', PCU1, R388, R6K, and RP4). The community composition and plasmid abundance were quantified by selective plating.

After the plasmids were introduced into the communities, dilutions of four different ratios ($10^3$, $10^4$, $10^5$, $10^6$) were

performed every 24 h to maintain the growth. We monitored the population dynamics daily over the following 15 days and obtained the fractions of each population and the plasmid (Supplementary Figs. 4b–h and 5c). We measured the conjugation rates and fitness costs (Supplementary Figs. 4i, j, 5a, b) and also estimated the plasmid loss rate. With these parameters, we determined the plasmid persistence potential in each community. The results were well matched to the predicted pattern, suggesting that $\omega$ values determine plasmid abundance in microbial communities (Fig. 4a).

To examine the general applicability of the metric, we reanalyzed the data from 9 previous studies that have provided sufficient measurements on the kinetic parameters and abundance of plasmids[17,36–43]. The microbial communities analyzed in these studies covered one, two, or three populations, and up to three plasmids. We collected a total of 83 data points (see Supplementary Information section 3) covering a $\omega$ range from $7.22 \times 10^{-4}$ to $2.62 \times 10^5$ (Fig. 4b). These data confirm the predictive power of $\omega$: in general, the plasmid persists when $\omega > 1$, and its relative abundance increases with $\omega$. In contrast, when $\omega < 1$, the plasmid tends not to persist, with its relative abundance close to 0.

## Discussion

Our work addresses two fundamental challenges facing the quantitative analysis of plasmid flow dynamics in microbial communities. First, it has been impractical to simulate a complex community in which many plasmids are transferred. This challenge is resolved by using our plasmid-centric modeling framework through drastic dimension reduction in model formulation. Second, the PCF enables the heuristic derivation of a metric that predicts the persistence and abundance of any plasmids in a microbial community based on its kinetic parameters and community composition. In the simplest communities of one species transferring only one plasmid, our persistence potential is

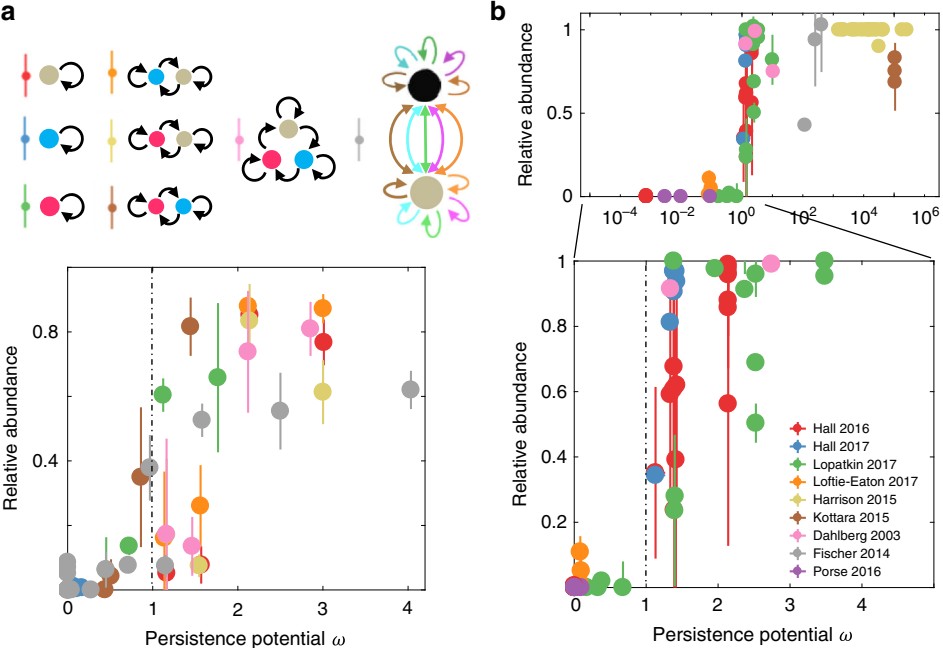

**Fig. 4 Experimental validation of the predictive power of $\omega$. a** Experimental test with eight synthetic communities. The first seven communities (upper panel) transferring a single plasmid were constructed using three engineered *E. coli* strains. Strain B (blue) expresses BFP chromosomally; strain R (red) expresses dTomato; and strain X (gray) does not express a fluorescent marker. We assembled three single-population communities, two pairwise communities, and one three-population community. Black arrows indicate the plasmid transfer. Dilutions of four different ratios ($10^3$, $10^4$, $10^5$, $10^6$) were performed every 24 h to maintain the growth. The relative abundances and persistence potentials of the plasmid in the seven communities at the end of the experiments (day 15) are shown in the lower panel. The error bars represent the standard deviations of three replicates. The eighth community transferring multiple plasmids was composed of two *E. coli* strains, MG1655 and DH5α. Five self-mobilizable plasmids, F' (IncF, Tet$^R$), PCU1 (IncN, Amp$^R$), R388 (IncW, Tm$^R$), R6K (IncX, Strp$^R$), and RP4 (IncP, Kan$^R$), were transferring within the community. The composition of the community was determined by blue-white screening on X-gal plates. The relative abundance of each plasmid was determined by selective plating with the corresponding antibiotics. Four different ratios ($10^3$, $10^4$, $10^5$, $10^6$) were performed every 24 h. The relative abundances and persistence potentials of the five plasmids at the end of the experiments (day 15) are shown in the lower panel. Data are presented as mean values $+/-$ the standard deviations of three biologically independent replicates. **b** Evaluation of literature data. Data were extracted and reanalyzed from 9 previous studies (Supplementary Tables 3–16). Data are presented as mean values $+/-$ the standard deviations of multiple biologically independent replicates. The number of replicates ranges from three to six, depending on the rationale of each study. The distribution is shown in the logarithmic scale (upper panel) or linear scale of $\omega$ (lower panel).

analogous to the criterion derived by Stewart and Levin[26]. However, our metric is generally applicable to communities composed of multiple species transferring multiple plasmids. Both the modeling framework and the derived plasmid persistence potential have implications for future efforts to understand, control, and exploit horizontal gene transfer dynamics in microbial communities.

For instance, the persistence potential $\omega$ predicts that the abundance of a plasmid in a microbial community is sensitive to the average growth rate of the constituent populations, especially when the growth rate is close to the system dilution rate. Consistent with this notion, nutrient enrichments, which in general increase the average growth rate of the community, have been shown to increase the relative abundance of plasmids[44]. In the inflamed mouse gut, the growth of commensal Enterobacteriaceae such as *E.coli* is boosted by the nitrate generated as a by-product of the host inflammatory response[45]; this transient boom of Enterobacteriaceae was shown to correlate with an increase in transconjugant abundance[46,47]. In both situations, an increase in the growth rate can promote persistence by increasing the persistence potential directly (through a decrease of its denominator term, Eq. (2)) or indirectly (through increasing the conjugation efficiency)[48]. Conversely, our metric suggests that modulating the overall growth rate of gut microbiota by nutrient supply or its dilution by water inflow could be effective strategies to regulate the abundance of the mobile gene pool in the human gut[49,50].

Past studies have suggested that species composition of a community influences the plasmid persistence. For instance, for bacteria with poor conjugation efficiencies, coculturing them with efficient donors has been shown to enhance plasmid transfer and maintenance[36,51]. However, the relationship between plasmid abundance and species composition have been qualitative. Here, our metric allows quantitative, albeit approximate, prediction of plasmid abundance given the composition and the kinetic parameters. The human gut contains an enormous diversity of microbes[52], and multiple factors such as diet[53], age[54], and antibiotic administration[55] can alter the composition of its microbiome. With more kinetic information required, our metric can enable a predictive mapping between such manipulations and plasmid abundance in gut. This notion will help to predict the effectiveness of diverse strategies of reversing antibiotic resistance in complex microbial communities[56].

Our results also have implications for the engineering of microbial consortia. Engineering complex bacterial communities in useful ways remains challenging due to the lack of understanding of the ecological principles and intercellular metabolic interactions[57]. Conjugative plasmids are potentially powerful tools for function-oriented microbiota engineering[58]. Our modeling framework and the plasmid persistence potential can guide such efforts. In particular, the persistence potential reveals how a few key kinetic parameters predict the approximate abundance of the plasmid.

Several caveats need to be considered when using our framework. Our framework does not account for the situation where the gene translocates from a plasmid to the chromosome or other plasmids[59], in which case the plasmid persistence becomes decoupled from the functions encoded by this gene. Our framework does not consider changes in gene functions during long-term evolution[17,25]. Moreover, our basic framework does not explicitly consider interactions between plasmids. However, certain interactions, such as plasmid incompatibility, can be described by PCF with minor modifications (see Supplementary Information section 2.1.2). Developed for plasmids, PCF can also be extended to analyze other MGEs, including phages and transposons, with major simplifications and adaptations (see Supplementary Information section 2.5). However, the dynamics of phages and transposons can entail different confounding factors[59–61]; thus, the quantitative predictive power of such a modeling framework remains to be thoroughly tested. Despite these caveats, PCF captures the general features of the population biology of plasmids, and the persistence potential established in this work is applicable when the genes of interest are stably associated with the plasmids and when the effects of evolution are negligible.

## Methods

**Strains and plasmids**. The compositions of the eight engineered communities are shown in Supplementary Table 1. *E.coli* strain MG1655 without fluorescence markers was denoted as strain X[17]. *E.coli* strain DA26735 with chromosomal BFP and chloramphenicol resistance (Cm^R) was denoted as strain B[17], and *E.coli* strain DA32838 with chromosomal dTomato and Cm^R was denoted as strain R[17]. All three strains carry plasmid helper F plasmid $F_{HR}$ that expresses tetracycline resistance (Tet^R)[17]. $F_{HR}$ is not transmissible but encodes the conjugation machinery that mobilize plasmid K. Plasmid K expresses GFP under the control of a strong constitutive PR promoter, and expresses kanamycin resistance (Kan^R)[17,62]. Plasmid K also carries *oriT*, so it can be transferred through conjugation[17].

The multi-plasmid community was composed of *E.coli* strain MG1655 and DH5α. These two strains were distinguished from each other via blue-white screening on X-gal plates, where MG1655 and DH5α colonies were blue and white, respectively. These communities transferred five conjugative plasmids: F' (lncF, Tet^R), PCU1 (lncN, Amp^R),R388 (lncW, Tm^R), R6K (lncX, Strp^R), and RP4 (lncP, Kan^R)[17]. These five plasmids are compatible with each other and carry different antibiotic resistance markers. The plasmids were distinguished from each other via selective plating.

**Long-term dynamics of the engineered microbial communities**. Our methods for measuring the long-term plasmid dynamics are based on the protocols established by Lopatkin et al.[17]. Single colonies of three strains (X, B and R) carrying plasmid $F_{HR}$ and K were grown overnight at 37 °C for 16 h with shaking (250 rpm) in LB culture (LB broth from APEX) containing appropriate antibiotics (100 μg/mL Cm, 50 μg/mL Kan, or 20 μg/mL Tet). The overnight cultures were resuspended in M9 medium (M9CA medium broth powder from Amresco, supplemented with 0.1 mg/mL thiamine, 2 mM MgSO₄, 0.1 mM CaCl₂, and 0.4% w/v glucose) without antibiotics, and diluted to the initial density of 10⁶ cells/mL. We constructed seven communities using the combinations of these three strains: (a) X; (b) B; (c) R; (d) X + B; (e) X + R; (f) B + R; (g) X + R + B. For the communities (d)–(g), the members of each community were mixed in the equal ratio, and the mixtures were diluted to the initial density of 10⁶ cells/mL. The cells were then distributed in a 96-well plate to a final volume of 200 μL/well, and each community had 12 replicates. The 96-well plate was covered with an AeraSeal^TM film sealant (Sigma-Aldrich, SKU A9224) followed by a Breath-Easy sealing membrane (Sigma-Aldrich, SKU Z380059). Plates were shaken at 250 rpm at 37 °C for 23 h. This was denoted as day 0. On day 1, the 12 replicates of each community were divided into four groups, each with three replicates. From each well, 2 μL was removed for flow cytometry. The four groups were subjected to four dilution ratios (10³×, 10⁴×, 10⁵×, 10⁶×). The new plates were sealed using both membranes and placed back into the incubator. The same protocols were repeated daily from day 1 to day 15.

For the community transferring multiple plasmids, we first transformed the five plasmids into MG1655, respectively, and obtained five plasmid-carrying strains (M^F, M^PCU1, M^R388, M^R6K, and M^RP4). Single colonies of the six strains (M^F, M^PCU1, M^R388, M^R6K, M^RP4, and DH5α) were grown overnight at 37 °C for 16 h with shaking (250 rpm) in LB culture containing appropriate antibiotics (20 μg/mL Tet, 100 μg/mL Amp, 10 μg/mL Tm, 50 μg/mL Strp, and 50 μg/mL Kan, respectively). The overnight cultures were resuspended in M9 medium without antibiotics and diluted to the initial density of 10⁶ cells/mL. The cells of these six strains were mixed in a ratio of 1:1:1:1:1:5 (M^F: M^PCU1: M^R388: M^R6K: M^RP4: DH5α), and the mixtures were diluted to the initial density of 10⁶ cells/mL.

The cells were then distributed in a 96-well plate to a final volume of 200 μL/well, with 12 replicates. The 96-well plate was covered with an AeraSeal^TM film sealant followed by a Breath-Easy sealing membrane. Plates were shaken at 250 rpm at 37 °C for 23 h. This was denoted as day 0. At day 1, the 12 replicates of each community were divided into four groups, each with three replicates. From each well, 10 μL was removed for selective plating. The four groups were subjected to four dilution ratios (10³×, 10⁴×, 10⁵×, 10⁶×), respectively. The new plates were sealed using both membranes and placed back to the incubator. The same protocols were repeated daily from day 1 to day 15. The ratio between MG1655 and DH5α cells was determined by plating on the X-gal plates (100 μg/mL X-gal, 1 mM IPTG). The relative abundance of each plasmid was determined by plating on the plates with the corresponding antibiotics (20 μg/mL Tet, 100 μg/mL Amp, 10 μg/mL Tm, 50 μg/mL Strp, and 50 μg/mL Kan).

**Flow cytometry**. The community composition and plasmid abundance were quantified using a flow cytometer (MACSQuant® VYB Analyzer). From day 1 to day 15, the overnight cultures were resuspended and diluted to 1: 1000 in 200 μL fresh M9 media before running through the flow. The channels of emission detectors were set as V1: CFP_VioBlue (450/50 nm) for BFP, Y2: dsRed_txRed (615/20 nm) for dTomato, and B1: GFP_FITC (525/50 nm) for GFP. For each sample, 10,000 cells were collected. All data analysis was performed using FlowJo (version 10.5.3).

**Measuring the fitness costs and conjugation efficiencies of the mobilizable plasmids**. Singles colonies of cells that carry the plasmid K (denoted X^K, B^K, and R^K) or do not carry the plasmid K (denoted X^0, B^0, and R^0) were grown overnight at 37 °C for 16 h with shaking in LB with appropriate antibiotics. The cultures were resuspended in M9 medium without antibiotics and then diluted in 1:10⁴ ratio. The growth curves of these six cell types were measured using a plate reader (TECAN infinite M200 PRO). Six replicates per cell type were used for quantification. The growth rate constants were calculated as the effective growth rates in exponential phases (e.g., the first 5 h). First, we smoothed the growth curves to filter out the random noises from the plate reader. We then plotted the increments $\Delta N/\Delta t$ with regards to the cell density $N$. Linear regression was performed, and the slope was obtained as the growth rate constants. The fitness cost $\alpha$ was determined by normalizing the growth rates of plasmid-free populations X^0, B^0, and R^0 by the growth rates of plasmid-carrying populations X^K, B^K, and R^K, respectively. The $\lambda$ values were obtained through $\lambda = \alpha - 1$. The fitness costs of the five conjugative plasmids (F', PCU1, R388, R6K, and RP4) in the host strain MG1655 were measured in the same way.

Conjugation efficiencies were estimated using the protocols established by Lopatkin et al. with modifications[17]. X^K, B^K, and R^K served as the donors, and X^0, B^0, and R^0 served as the transconjugants. To distinguish donors with transconjugants, we transformed the recipients with another non-mobilizable plasmid pJM31, which carried ColE1 origin and expressed ampicillin resistance (Amp^R). Therefore, donors, recipients, and transconjugants can be distinguished by different resistance markers: Kan for donors, Amp for recipients, and Kan +Amp for transconjugants. Overnight cultures of donors and recipients in LB media with appropriate selection agents were resuspended and diluted (1:100) in fresh LB media. Cells were incubated at 37 °C with shaking for 2 h until they reached exponential phase. The cells were then resuspended in M9 media and mixed in 1:1 ratio with a total volume of 200 μL. Mixtures were incubated at room temperature (25 °C) for 1 h without shaking. The donor, recipient, and transconjugant densities were measured by diluting the mixtures (1:10⁶ for donor and recipient, 1:10⁴ for transconjugant) and spreading three replicates onto corresponding selective plates. The conjugation efficiency was obtained as $\eta = \frac{T}{D \cdot R \cdot \Delta t}$, where $T$, $D$, $R$ stand for the cell densities of transconjugant, donor and recipient, respectively. Before being plugged into the model, the measured values of $\eta$ need to be normalized with respect to the maximum carrying capacity $N_m$. We estimated $N_m$ to be $6 \times 10^8$ cells/mL, which corresponds to OD600 ≈ 1.2. The dilution rates $D$ were calculated from the daily dilution ratio $\varepsilon$ by $D = \frac{\log \varepsilon}{24 h}$[63]. Therefore, dilution ratios of 10³×, 10⁴×, 10⁵×, 10⁶× are equal to dilution rates of 0.2878 h⁻¹, 0.3838 h⁻¹, 0.4797 h⁻¹, and 0.5756 h⁻¹, respectively. The loss rate $\kappa$ of plasmid K is very small compared with dilution rates $D$. We used the value ($\kappa = 0.001\ h^{-1}$) measured by Lopatkin et al.[17].

The conjugation efficiency of the five conjugative plasmids were measured using similar protocols. Since the abundance of DH5α became negligible in the cocultures at the end of the long-term experiments, only the conjugation between MG1655 cells were considered. For plasmid F' (Tet^R), PCU1 (Amp^R),R388 (Tm^R), and R6K (Strp^R), MG1655 transformed with the non-mobilizable plasmid K- (Kan^R, same as the plasmid K but without the transfer origin) served as the recipient. For plasmid RP4 (Kan^R), MG1655 transformed with the non-mobilizable plasmid pJM31 (Amp^R) served as the recipient. The calculations of parameters were performed using Matlab (R2017a).

**Model construction and analysis**. To explain the key concepts of the plasmid-centric framework, we first focus on simple communities. For a community of two species and two plasmids, let $s_1$ and $s_2$ represent the abundances of species 1 and 2, respectively. Let $p_{11}$ represent the abundance of species-1 cells that carry plasmid 1,

and $p_{12}$ represent the abundance of species-1 cells that carry plasmid 2. In a similar way, we can define $p_{21}$ and $p_{22}$. PCF describes how the community composition ($s_i$, $i = 1$, 2) and plasmid distribution ($p_{ij}$, $i = 1$, 2 and $j = 1$, 2) change with time. First, we assume all plasmids are compatible with each other, which means that they can coexist in the same host cell. The dynamics of this community can then be described by six ODEs

$$\frac{ds_1}{dt} = \alpha_1 \mu_1^e s_1 - D s_1 \tag{5}$$

$$\frac{ds_2}{dt} = \alpha_2 \mu_2^e s_2 - D s_2 \tag{6}$$

$$\frac{dp_{11}}{dt} = \beta_{11} \mu_{11}^e p_{11} + (s_1 - p_{11})(\eta_{111} p_{11} + \eta_{121} p_{21}) - (\kappa_{11} + D) p_{11} \tag{7}$$

$$\frac{dp_{12}}{dt} = \beta_{12} \mu_{12}^e p_{12} + (s_1 - p_{12})(\eta_{211} p_{12} + \eta_{221} p_{22}) - (\kappa_{12} + D) p_{12} \tag{8}$$

$$\frac{dp_{21}}{dt} = \beta_{21} \mu_{21}^e p_{21} + (s_2 - p_{21})(\eta_{122} p_{21} + \eta_{112} p_{11}) - (\kappa_{21} + D) p_{21} \tag{9}$$

$$\frac{dp_{22}}{dt} = \beta_{22} \mu_{22}^e p_{22} + (s_2 - p_{22})(\eta_{222} p_{22} + \eta_{212} p_{12}) - (\kappa_{22} + D) p_{22}. \tag{10}$$

These ODEs constitute the main part of PCF. $s_i$ increases by cell division, the effective growth rate of which is represented by $\mu_i^e$. Each plasmid might cause a fitness burden or benefit on the growth of the host cell, and $\alpha_i$ represents the combined fitness of all the plasmids that species $i$ carries. $D$ is the dilution rate. $p_{ij}$ increases by cell division ($\mu_{ij}^e$) or horizontal transfer through conjugation ($\eta_{jki}$). The effective growth rate of the host cells that carry plasmid $j$ is represented by $\mu_{ij}^e$, which will be smaller than $\mu_i^e$ if the plasmid is burdensome and larger than $\mu_i^e$ if the plasmid brings benefit. The combined effect of all other plasmids on the division rates of $p_{ij}$ is described by $\beta_{ij}$. Plasmid $j$ can also be transferred horizontally from species $k$ to species $i$ at a rate constant of $\eta_{jki}$. Therefore, the influx of $p_{ij}$ from species $k$ is obtained as $(s_i - p_{ij})\eta_{jki} p_{kj}$. The rate constant of plasmid loss is represented by $\kappa_{ij}$.

$\mu_i^e$ represents the effective growth rate of the 'empty' cells (cells that do not carry any plasmids), while $\mu_{ij}^e$ is the effective growth rate of the cells that are only equipped with the plasmid $j$. The effective growth rates $\mu_i^e$ and $\mu_{ij}^e$ are calculated from the maximum growth rates (represented by $\mu_i$ and $\mu_{ij}$, respectively) and the available carrying capacity $c_i$ through $\mu_i^e = \mu_i c_i$, $\mu_{ij}^e = \mu_{ij} c_i$. $\mu_{ij}$ is linked to $\mu_i$ through the fitness cost of $p_{ij}$, which is denoted $\lambda_{ij}$. We assume that the maximum growth rate is inversely proportional to the fitness cost. Therefore, the relationship between $\mu_{ij}$ and $\mu_i$ is obtained as $\mu_{ij} = \mu_i/(1 + \lambda_{ij})$. The plasmid is burdensome with positive $\lambda_{ij}$ and beneficial with negative $\lambda_{ij}$. To obtain the combined cost of all the plasmids that $s_i$ carries, we calculated the weighted average of their costs as $\overline{\lambda_i} = \frac{p_{i1}}{s_i}\lambda_{i1} + \frac{p_{i2}}{s_i}\lambda_{i2}$. Then, the formulation of $\alpha_i$ can be obtained as $\alpha_i = \frac{1}{1+\overline{\lambda_i}}$, which leads to

$$\alpha_1 = \frac{s_1}{s_1 + p_{11}\lambda_{11} + p_{12}\lambda_{12}} \tag{11}$$

$$\alpha_2 = \frac{s_2}{s_2 + p_{21}\lambda_{21} + p_{22}\lambda_{22}}. \tag{12}$$

Formulating $\beta_{ij}$, however, requires prior information about the distribution patterns of the other plasmids in $p_{ij}$ cells. For a general understanding, we equilibrated its distribution in $p_{ij}$ cells as its distribution in $s_i$ cells. Then, similar to the definition of $\alpha_i$, $\beta_{ij}$ can be obtained as

$$\beta_{11} = \frac{s_1(1 + \lambda_{11})}{s_1(1 + \lambda_{11}) + p_{12}\lambda_{12}} \tag{13}$$

$$\beta_{12} = \frac{s_1(1 + \lambda_{12})}{s_1(1 + \lambda_{12}) + p_{11}\lambda_{11}} \tag{14}$$

$$\beta_{21} = \frac{s_2(1 + \lambda_{21})}{s_2(1 + \lambda_{21}) + p_{22}\lambda_{22}} \tag{15}$$

$$\beta_{22} = \frac{s_2(1 + \lambda_{22})}{s_2(1 + \lambda_{22}) + p_{21}\lambda_{21}}. \tag{16}$$

Here, we assume the two species compete with each other and follow the logistic growth. Therefore, the available carrying capacities can be formulated as $c_1 = c_2 = 1 - s_1 - s_2$.

The framework can then be generalized to communities with more species and plasmids. First, we discuss communities composed of $m$ species and $n$ plasmids. Let $s_i$ ($i = 1, 2, \ldots, m$) represent the abundance of species $i$, and $p_{ij}$ ($i = 1, 2, \ldots, m$ and $j = 1, 2, \ldots, n$) represent the abundance of plasmid $j$-carrying cells in species $i$. We assume all the plasmids are compatible with each other. The dynamics of this

community can then be described as follows:

$$\frac{ds_i}{dt} = \alpha_i \mu_i s_i c_i - D s_i \tag{17}$$

$$\frac{dp_{ij}}{dt} = \beta_{ij} \mu_{ij} p_{ij} c_i + (s_i - p_{ij}) \sum_{k=1}^{m} \eta_{jki} p_{kj} - (\kappa_{ij} + D) p_{ij}. \tag{18}$$

The formulation of $\alpha_i$ in this general case becomes

$$\alpha_i = \frac{s_i}{s_i + \sum_{j=1}^{n}(p_{ij}\lambda_{ij})} \tag{19}$$

and $\beta_{ij}$ becomes

$$\beta_{ij} = \frac{s_i(1 + \lambda_{ij})}{s_i(1 + \lambda_{ij}) + \sum_{\{k:k\neq j\}}(p_{ik}\lambda_{ik})}. \tag{20}$$

The ODEs, and the formulations of $\alpha$, $\beta$, and $c$, constitute the body of our framework.

**Niche-based simulation**. The term $c_i$ represents the available carrying capacity of the species. Here, we assume each species follows logistic growth, in competition with other species sharing the same niche. Let $e_i$ represent the maximum carrying capacity of the niche where the $i$-th species resides. We formulated $c_i$ as $c_i = e_i - \sum s_k$, where $s_k$ is the density of the $k$-th species that shares the same niche.

To numerically test the predictive power of the persistence potential, we first randomized the number of species and plasmids in each community. We then generated a random number of the niches and distributed each species into one of the niches. Each niche was assigned a random value of the maximum carrying capacity. Next, the maximum rates of the species and the dilution rate were randomized. For plasmid dynamics, the transfer rates of each plasmid were randomized in two steps: (1) the transfer rate between a specific pair of species might be zero or no-zero, by a random chance; (2) if the transfer rate is non-zero, the value of $\eta$ is also randomized. Finally, the plasmid loss rates, the growth effects, as well as the initial densities of each population were randomized. With all these parameters, we then simulated the dynamics of the communities. All the parameters follow uniform distributions in the given ranges. The pipeline of the simulation and the ranges of the parameters were described in detail in Section 2.1.3 of the Supplementary Information. The numerical simulations were performed using Matlab (R2017a).

**Reporting summary**. Further information on research design is available in the Nature Research Reporting Summary linked to this article.

## Data availability
The authors declare that all data of this study are available within the manuscript and its Supplementary Information file. The summary of the literature data is provided in Supplementary Tables 3–16. Source data associated with Fig. 4, Supplementary Figs. 4 and 5 are provided with this paper. Any additional information is available upon request.

## Code availability
The Matlab codes associated with Figs. 1 and 3, and Supplementary Figs. 1–3 are available at the Github repository (https://github.com/youlab/PlasmidPersistencePotential_TengWang).

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

## Acknowledgements

We thank Allison J. Lopatkin, Cheemeng Tan, Hong Qian, Terry Hwa, and Lawrence David for thorough reading and comments on an earlier draft of the manuscript and James P. Hall for kindly providing access to the original data of his published work. We thank Ryan Tsoi, Helena Ma, Feilun Wu, Nan Luo, Jia Lu, Zach Holmes, Katherine Duncker, and Andrea Weiss for comments on the manuscript. We also thank Nicolas Buchler for access to the flow cytometer and Firas Midani for flow cytometry trouble shooting. This work is partially supported by the National Institutes of Health (L.Y., and R01A1125604 and R01GM110494) and David and Lucile Packard Foundation (L.Y.).

The funders had no role in study design, data collection and analysis, decision to publish, or preparation of the manuscript.

## Author contributions

T.W. conceived the research, designed, and performed both modeling and experimental analyses, interpreted the results, and wrote the manuscript. L.Y. conceived the research and assisted in research design, data interpretation, and manuscript writing.

## Competing interests

The authors declare no competing interests.
