## [Peer Review File · Nature Communications]

REVIEWER COMMENTS

Reviewer #1 (Remarks to the Author):

This paper describes an alternative model for tracking the spread of mobile genetic elements (MGEs) through a population. This simplifies the standard modelling approach by considering the total abundance of the strain population carrying a certain MGE instead of tracing each sub-population of the strain with different MGE combinations. They derive a threshold metric for determining persistence of MGEs which they compare to experimental data (their own and from the literature).

We believe the model structure and metric developed are valuable, that the paper is generally sound and mostly have interpretation comments that we would like addressed before it is published.

Major comments

Plasmids are a very important mobile genetic element. We believe that this model should be described as applying primarily to plasmids and conjugation throughout. With future work being the adaptation, with the need for more data, of its application to transposons and lytic phage. Indeed, the idea of persistence we would argue is most important for plasmids and their encoded genetic material. For example, when / why would it be important to consider the persistence of a lytic bacteriophage?

Following on from the above comment, we believe care should be taken throughout to distinguish between persistence of the MGE and of its associated genetic material. It is not the plasmid per se that is required to remain in the bacterial host but the gene encoding the desired characteristic. (e.g. throughout Introduction / end of Results)

Is there open source code available to support this analysis and the results?

The discussion section is rather weak: Where are the limitations? Where is the comparison to other modelling work? What are the next steps?

How important are the discrepancies between the MCF and the SCF? They seem to only be at extreme fitness values? Could you highlight this more as a strength whilst discussing where the weaknesses are i.e. the parameter space in which using this simplified model breaks down? Is it up to a 20% discrepancy?

The implications of the work weren't quite clear. What of the components of the threshold measure can actually be edited to change MGE persistence?

Introduction

I think some care is needed to differentiate between genes and MGEs: for example transposon carrying an antibiotic resistance gene could be located on a plasmid. The important thing is the persistence of the gene encoding the desired (or not wanted) trait.

Will this work really propose a strategy to overcome the issue of promoting MGE persistence? How does it?

"Currently impossible" to model: I would argue that it is computationally expensive, but not impossible. Many of these parameters are also needed for the model proposed.

Results

It needs checking whether this model construction / comparison is really a result under the journal's requirements.

Expand ODE (would argue not a standard nomenclature)

Equations 1, 2 and 3: please remove the unnecessary comma at the end of each line

What is "the dilution rate"?

How do you calculate the "combined growth effect"? is this like a relative fitness term? Would the latter be a better link to the literature?

Can you make it clear you are only looking at plasmids as an example?

2nd paragraph: I would make it clear here that $\mu_i^e = \mu_i c_i = \mu_i (1 - \sum_k s_k)$, i.e. that $\mu^e \neq \mu$. I know that you explain this in the Supplementary Information, but I think this addition

would help the reader immediately understand that you assume logistic growth, and prevent any confusion as to what μ is in Equation 3.

Paragraph 5: Replace "lumping" with "combining"?

Paragraph 6: I recommend deleting the sentence stating that lytic phage can be described by the MCF (please refer to Major comments for details).

Paragraph 6: I recommend editing the sentence on transposons, as I don't think that the MCF is directly applicable to transposons if they are transferred by transduction (please refer to Major comments for details)

Paragraph 8: Please clearly state what λ is

Paragraph 8: what happened if $0 < w < 1$?

Paragraph 9: Does the model need the population to be at equilibrium? Is this realistic?

Paragraph 10: 30,000 hours corresponds to 3 years; is it realistic to assume that all the parameter values (bacterial growth rate, cost/benefit of MGE, horizontal transfer rate etc...) would remain constant for such a long time? I understand this is done to reach a steady state, but it would be good to briefly discuss this.

Discussion

As mentioned above, I think that this model is not refined enough to be any MGE yet. This framework could be used for any but not yet.

2nd paragraph: Is this result in the results section?

3rd paragraph: isn't that result to be completely expected?

3rd paragraph: what are the new insights into reversing antibiotic resistance? Is it possible to target antibiotics at more efficient conjugators?

4th paragraph: What are the few key kinetic parameters that we have any influence over? Fitness cost of plasmid? Dilution rate?

Figure 3:

(C) Can you say that it is a robust predictor here with no data?

What are the niches? How big are they?

Figure 4:

Can you add a vertical line for $w = 1$ to aid in the importance of this as a threshold

Methods

Where are the methods for the model construction / analysis / code?

Supplementary Information

Equations 23 and 24: you mention a q parameter just below these equations, but that term is not in the Equations. I suggest you double-check all equations in case there are other minor typos hidden there.

Reviewer #2 (Remarks to the Author):

In the paper "The Persistence Potential of Mobile Genetic Elements" the authors propose a theoretical framework to evaluate the stability of plasmids in microbial communities. As the authors argue, this is an important problem that has been studied since the seminal studies by Levin and Stewart in the 1970s. In this paper, the authors argue that the subpopulation-centered framework used previously to study the population dynamics of plasmid-bearing populations has many limitations and thus cannot be used to study complex microbial communities. This statement is correct and, for this reason, several generations of mathematical modelers have restrained from using ODE models to study microbial communities composed of multiple strains and plasmids. To

circumvent this problem, the authors propose to use a so-called plasmid-centric framework, a modeling approach that describes explicitly the relative abundances of MGEs in the community from the average fitness effects to the host.

This is a potentially interesting approach that reduces the number of equations and parameters needed to model diverse communities with multiple plasmids transferring vertically and horizontally. For instance, the authors make a back-of-the-envelope calculation that, in order to model the microbiome in a bottle of water, it would be necessary to use 10^{56} equations and 10^{114} parameters using the standard modeling framework, while using their approach it would *only* be necessary to use 28,640 equations and "a few kinetic parameters": 5.2×10^6 parameters values. I strongly agree with the authors that using ODEs to model complex microbial communities is a futile endeavor, but I would argue that this statement also holds for their proposed framework. Indeed, solving non-linear ODE systems with millions of unknown and potentially unidentifiable parameters may be possible from a computational perspective nowadays, but evaluating the robustness of the results to parametric and modeling uncertainties would be absolutely impossible. To numerically solve their model, the authors make a series of simplifying assumptions (not all stated clearly in the main text), for instance that competition occurs in niches (therefore effectively reducing the size of the community) and that plasmid burden can be estimated from a weighted average of their costs (therefore ignoring pleiotropic effects between plasmids). Moreover, parameters were randomized (although no details are given about the distribution from where parameters are sampled) to perform 2,000 simulations of 5-100 species and 1-50 plasmids for 30,000 hours. The authors argue that both approaches yield similar results.

Finally, to evaluate the heuristic expression derived from their model, the authors engineer eight synthetic communities carrying a mobilizable fluorescent plasmid. By performing serial dilution experiments for 15 days, the authors estimated conjugation rates and fitness costs, as well as segregational loss rates for each plasmid-strain association. From these parameters, the authors estimated the persistence potential and conclude that it is in good agreement with experimental data. This expression was also validated using data obtained from the literature. I find it surprising that the authors use simple experimental model systems to validate their approach, given that the potential benefit of using an MGE-centric approach would only be clear in large communities. Furthermore, to parametrize the model and estimate the persistence potential of MGEs, the authors need to estimate the fitness effects of each plasmid to each strain, an impossible task for complex microbial communities like soil or the microbiome but feasible for small communities like those under consideration. For small communities like the ones tested experimentally, however, the conventional subpopulation-centric approach could be used to determine the persistence potential of plasmids - as has been done for decades.

In summary, this manuscript addresses an important and significant problem in plasmid biology. However, this is a very difficult paper to follow, as most of the information necessary to understand their model and experimental systems can only be found in the Supplementary Information. Also, introducing new models and biological systems (i.e. phages and transposons) at the end of the manuscript makes it very difficult to understand. And while the proposed MGE-centric framework proposed is potentially useful, I believe it is more appropriate for a theoretical journal where the limitations of this modeling approach would be better scrutinized and its benefits better appreciated.

Reviewer #3 (Remarks to the Author):

The authors seek to find a solution for the problem of modelling many mobile genetic elements in bacterial populations, and by defining a new frame work in which mobile elements and bacterial

communities are separated in different ODE's they can reduce the number of ODE's enormously. This is an important addition for theoretical work in which both MGE dynamics and bacterial community dynamics are of interest, such as microbiome of the gut and antimicrobial resistance. The authors derive a persistence potential, and from this they show based on historical data that their modelling framework can determine the fate of MGE's.

This work is a continuation on the work of Stewart & Levin, Simonsen et al, and Lopatkin et al. extending it to multi-MGE microbial communities.

There are a few suggestions I would do before further publication:

- The biological definition of the persistence potential is unclear. Especially in line 137-138 it gets confusing as $\omega < 0$ also indicates persistence, but a prevalence below 0 seems unreasonable. Furthermore the graph Fig 3 C does not go below 0.

Considering the mathematics it seems to be the value for which a non-negative equilibrium exists. Thus it would have a similar interpretation as the basic reproduction number in ecology and epidemiology. If so it would be a good idea if the authors place the ideas in the context of such widely used concepts. Not necessarily but also of interest might be to consider percolation thresholds such used by Davis et al Nature volume 454, pages 634-637 (2008)

- The derivation of the community with diversity lines 146-149 and the supplementary information do not give any argumentation or proof that this is correct. If it is a pragmatic choice that worked out properly it is fine, but then please indicate so.

- The definition of α , μ , λ etc needs to be given in the main text. Also the fact that α and λ depend on p_{ij} and s_i must be part of the main text. Otherwise the model cannot be understood from the main text.

Minor:

Line 35-37: The list of environments is not exhaustive and should be indicated as such. Furthermore addition systems of plasmids are missing.

Line 73: Unclear what is exactly meant by "mobilizable" plasmids.

Line 74: Dimensions of abundance are not given. Later on s_i is given as relative abundance.

In Figure 4 B, two estimates with $\omega \approx 10^{-2}$ show a relative abundance above 1, but this is not discussed.

- Line 420: Why is this method to determine the conjugation efficiency used and not the end point method?

Kind regards
Egil Fischer

Point-by-point responses (in black) to reviewers' comments (in blue)

Reviewer #1:

Overview

This paper describes an alternative model for tracking the spread of mobile genetic elements (MGEs) through a population. This simplifies the standard modelling approach by considering the total abundance of the strain population carrying a certain MGE instead of tracing each sub-population of the strain with different MGE combinations. They derive a threshold metric for determining persistence of MGEs which they compare to experimental data (their own and from the literature). We believe the model structure and metric developed are valuable, that the paper is generally sound and mostly have interpretation comments that we would like addressed before it is published

We thank the reviewer for recognizing the value of our work. We also appreciate the reviewer's comments on the interpretation aspects of our manuscripts. We have fully addressed the raised issues below and in our revised manuscript.

Issues

1. Plasmids are a very important mobile genetic element. We believe that this model should be described as applying primarily to plasmids and conjugation throughout. With future work being the adaptation, with the need for more data, of its application to transposons and lytic phage. Indeed, the idea of persistence we would argue is most important for plasmids and their encoded genetic material. For example, when / why would it be important to consider the persistence of a lytic bacteriophage?

We thank the reviewer for the insightful suggestion. We agree that more data are required to demonstrate the application of our model to phages and transposons. In light of this suggestion, we have narrowed the scope of the work to focus on plasmid conjugation. As a result, we have updated the manuscript throughout to reflect this change.

2. Following on from the above comment, we believe care should be taken throughout to distinguish between persistence of the MGE and of its associated genetic material. It is not the plasmid per se that is required to remain in the bacterial host but the gene encoding the desired characteristic. (e.g. throughout Introduction / end of Results)

The reviewer raised an important point. Indeed, the persistence of a plasmid doesn't necessarily mean the persistence of an associated gene, if the gene can jump away (e.g. to another plasmid that's subsequently lost). However, the persistence of the plasmid indeed reflects that of the associated gene that cannot jump (or before it jumps). We have revised our manuscript to be more precise throughout the main text, to avoid any confusion.

3. Is there open source code available to support this analysis and the results?

We appreciate the reviewer for the suggestion. We created a Github repository for our codes and stated "The Matlab codes are available at the Github repository (https://github.com/youlab/PlasmidPersistencePotential_TengWang)" in Code Availability.

4. The discussion section is rather weak: Where are the limitations? Where is the comparison to other modelling work? What are the next steps?

We appreciate the reviewer for the comment. We agree that an in-depth discussion on the constraints of this framework is necessary. In the revised manuscript, we added one paragraph into the Discussion to address the raised issue:

- “Several caveats need to be considered when using our framework. Our framework does not account for the situation where the gene translocates from a plasmid to the chromosome or other plasmids¹, in which case the plasmid persistence becomes decoupled from the functions encoded by this gene. Our framework does not consider changes in gene functions during long-term evolution^{2,3}. Moreover, our basic framework does not explicitly consider interactions between plasmids. However, certain interactions, such as plasmid incompatibility, can be described by PCF with minor modifications (see *Supplementary Information* section 2.1.2). Developed for plasmids, PCF can also be extended to analyze other MGEs, including phages and transposons, with major simplifications and adaptations (see *Supplementary Information* section 2.5). However, the dynamics of phages and transposons can entail different confounding factors^{1,4,5}; thus, the quantitative predictive power of such a modeling framework remains to be thoroughly tested. Despite these caveats, PCF captures the general features of the population biology of plasmids, and the persistence potential established in this work is applicable when the genes of interest are stably associated with the plasmids and when the effects of evolution are negligible.”

5. How important are the discrepancies between the MCF and the SCF? They seem to only be at extreme fitness values? Could you highlight this more as a strength whilst discussing where the weaknesses are i.e. the parameter space in which using this simplified model breaks down?

We thank the reviewer for the comment and the suggestion. Indeed, the reviewer’s interpretation is consistent with our results.

In our framework, we combined multiple distinct subpopulations into an average one and then calculated the average fitness effect of the plasmids. We analyzed the discrepancies to evaluate whether this averaging strategy introduce significant bias to the simulation results. As shown in Fig. S2, in a wide range of fitness effects, from $\lambda = -0.5$ (the plasmid-carrying cells grow twice as fast as plasmid-free cells) to $\lambda = 1$ (the plasmid-carrying cells grow 50% slower than the plasmid-free), the discrepancies were very small. The discrepancies are only significant at extreme fitness values. These results suggest that our framework was a good approximation of complex plasmid dynamics.

To highlight this as a strength, we stated “In general, we note that both frameworks represent approximations of real biological systems. The ultimate test of each framework should be from experiments”. As suggested by the reviewer, we discussed the weakness of our framework in Discussion in greater details.

6. Is it up to a 20% discrepancy?

We thank the reviewer for the comment. Our simulation results suggested that, at extreme fitness values, the discrepancy can exceed 20%. For instance, with the extreme burden of $\lambda = 10$, the discrepancy between these two frameworks can reach 50%. In general, the plasmid-centric framework represents a good approximation of the population-centric framework when λ is small or moderate.

7. The implications of the work weren’t quite clear. What of the components of the threshold measure can actually be edited to change MGE persistence?

We appreciate the reviewer for the feedback. As shown in Fig. 3A, plasmid transfer efficiency, species growth rates, plasmid fitness cost, segregation loss, dilution rate and community composition can be modulated to change the persistence potential. We have updated the text to further clarify this point.

8. I think some care is needed to differentiate between genes and MGEs: for example, transposon carrying an antibiotic resistance gene could be located on a plasmid. The important thing is the persistence of the gene encoding the desired (or not wanted) trait.

We thank the reviewer for raising this insightful point. We agree that there is difference between persistence of the MGE and that of an associated gene, if the gene can jump off the MGE.

We have revised our manuscript to be more precise throughout the main text.

9. Will this work really propose a strategy to overcome the issue of promoting MGE persistence? How does it?

We appreciate the reviewer for the comment. Our results suggested that plasmid persistence potential can be promoted by increasing plasmid transfer efficiency and species growth rates, or reducing plasmid fitness cost, segregation loss or dilution rate. Modulating community composition can also change the persistence of the plasmids. The persistence potential provides a quantitative threshold of to what magnitude each parameter needs to be changed to eliminate or maintain the plasmids.

10. Currently impossible' to model: I would argue that it is computationally expensive, but not impossible. Many of these parameters are also needed for the model proposed

We appreciate the reviewer for the comment.

Indeed, we agree that future development of computing technology may make possible the computation that is *currently impossible*.

To illustrate the computational challenge, consider a community of 160 species transferring 180 plasmids, which require $\sim 2.4 \times 10^{56}$ ODEs if we use the SCF. With such huge complexity, it is practically impossible to construct or simulate such a model using the current computing technology. For example, let's assume we need 100 bytes to encode each equation (likely an underestimate). We would need $\sim 2.4 \times 10^{58}$ bytes just to store the equations. In 2011, it was estimated that the total computer storage in the world was ~ 300 exabytes (i.e., 300×10^{18} bytes) and had been increasing at 23% per year⁶. Thus, even the storage requirement of such a model will far exceed the current, combined capability of the entire world.

In light of the reviewer's comment, we have revised the text to be more precise.

11. It needs checking whether this model construction/comparison is really a result under the journal's requirements

We appreciate the reviewer for the suggestion. In the updated manuscript, we have made sure that the formats of the model are consistent with the journal's requirements.

12. Expand ODE (would argue not a standard nomenclature)

We thank the reviewer for the suggestion. When we first introduce the term ODE, we define it with the expanded term: "In SCF, a population carrying a particular combination of plasmids is considered a unique subpopulation that requires one **ordinary differential equation (ODE)** to describe".

13. Equations 1, 2 and 3: please remove the unnecessary comma at the end of each line

We appreciate the reviewer for thorough reading and the suggestion. In the revised manuscript, we removed the comma in Equations 1, 2, 3 and 4. We made the same changes for all the equations in the main text and the supplementary information.

14. What is ‘the dilution rate’?

We thank the reviewer for pointing it out. As with the previous works, we considered a microbial community as an open system where flows enter the community and leave the community with resources, waste and cells. The dilution rate D is the rate of the flow through the habitat as measured in turnovers per hour.

We have revised the main text to clarify this term.

15. How do you calculate the ‘combined growth effect’? is this like a relative fitness term? Would the latter be a better link to the literature?

We thank the reviewer for the question. If we use the growth rate as the metric of the population fitness, the “combined growth effect” corresponds to the relative fitness. We did consider using “fitness” instead of “growth rate”. One potential caveat we see is that fitness can have different interpretations depending on the context or how experiments/calculations are done. For instance, even when describing a clonal microbial population, fitness can be interpreted as the maximal growth rate or the final density. As such, we prefer to use terms like “growth rate” or “growth effect” to avoid potential ambiguity.

In light of the reviewer’s comment, we have further clarified the meaning of “combined growth effect” and how it is calculated.

16. Can you make it clear you are only looking at plasmids as an example?

We thank the reviewer for the suggestion. In the updated version, we focused on the persistence of plasmids only.

17. 2nd paragraph: I would make it clear here that $\mu_i^e = \mu_i c_i = \mu_i (1 - \sum_k s_k)$, i.e. that $\mu^e \neq \mu$. I know that you explain this in the Supplementary Information, but I think this addition would help the reader immediately understand that you assume logistic growth, and prevent any confusion as to what μ is in Equation 3.

We appreciate the reviewer for the suggestion. We have revised the main text to clarify this distinction.

18. Paragraph 5: Replace ‘lumping’ with ‘combining’?

We thank the reviewer for the suggestion. We have revised the text accordingly.

19. Paragraph 6: I recommend deleting the sentence stating that lytic phage can be described by the MCF (please refer to Major comments for details).

Paragraph 6: I recommend editing the sentence on transposons, as I don’t think that the MCF is directly applicable to transposons if they are transferred by transduction (please refer to Major comments for details).

We appreciate the reviewer for the suggestions. We removed the sentences of applying our framework to phages and transposons. In the revised manuscript, we focused on the plasmid dynamics only.

20. Paragraph 8: Please clearly state what λ is

We thank the reviewer for the suggestion. We have revised the text to clarify the definition of λ . It reflects the burden of a plasmid.

21. Paragraph 8: what happened if $0 < \omega < 1$?

If $0 < \omega < 1$, the plasmid is predicted to be lost in the simplified model (where all populations are identical to each other). In general, a community may deviate from the criterion (e.g. see Figure.3c). Thus, in general, if $0 < \omega < 1$, the plasmid will be in low abundance or lost.

We have revised the text to be precise when describing the criterion.

22. Paragraph 9: Does the model need the population to be at equilibrium? Is this realistic?

Paragraph 10: 30,000 hours corresponds to 3 years; is it realistic to assume that all the parameter values (bacterial growth rate, cost/benefit of MGE, horizontal transfer rate etc...) would remain constant for such a long time? I understand this is done to reach a steady state, but it would be good to briefly discuss this

We thank the reviewer for the comment. When deriving the criterion from the simple model, we assume the system is at steady state. When considering complex communities (Figure.3c), we run the simulations long enough to approximate the steady state as it cannot be solved analytically. However, as illustrated by additional numerical simulations, the criterion maintains its predictive power even when the system is still far from the steady state (Supplemental Figures. S3).

We have revised the text to further clarify this point.

23. As mentioned above, I think that this model is not refined enough to be any MGE yet. This framework could be used for any but not yet. 2nd paragraph: Is this result in the results section?

We thank the reviewer for the comment. We agree that at the current stage we should limit our discussion in plasmids. We updated the manuscript to reflect this change.

The second paragraph discusses the implication of our criterion. It's not part of our results. We have revised the text to make this point clear.

24. 3rd paragraph: isn't that result to be completely expected?

3rd paragraph: what are the new insights into reversing antibiotic resistance? Is it possible to target antibiotics at more efficient conjugators?

We thank the reviewer for the comments. Indeed, we agree that it is not surprising to expect that the composition of a community would influence the plasmid persistence. However, how and to what extent this influence occurs is not evident. What our criterion reveals is *quantitatively* how different factors associated with the plasmids and the community composition affect plasmid persistence. This quantitative relationship was not evident.

If an antibiotic gene is associated with a plasmid of interest, any factor (or combinations of factors) that reduces plasmid persistence can facilitate reversal of antibiotic resistance. According to our criterion, this can occur if we reduce the average growth rate of different populations, reduce the conjugation efficiency, increase the plasmid burden, or increase the dilution rate. In theory, it is indeed possible to use chemicals (including antibiotics) that can target conjugation. However, we are not aware of any current antibiotics that can also effectively reduce conjugation.

25. 4th paragraph: What are the few key kinetic parameters that we have any influence over? Fitness cost of plasmid? Dilution rate?

Each of the kinetic parameters in the criterion (Eq.4 in the main text) can be controlled experimentally, to varying degrees. For instance, the average growth rate and the dilution rate can be readily changed. Plasmid loss or conjugation inhibition can be promoted by certain chemicals. The fitness cost of a plasmid can be changed by introducing a chemical that induce high-level gene expression off the plasmid. However, we recognize that not all of these strategies are applicable in dealing with natural communities. As such, we have revised our discussion to focus on the strategies that can operate on natural communities.

26. Figure 3: (C) Can you say that it is a robust predictor here with no data?

We appreciate the reviewer for the comment. By “robust”, we meant to compare the simple criterion (Eq. 3) with the simulation results from complex communities. Thus, the comparison is with simulated data.

In light of the reviewer’s comment, we have revised the figure caption and the corresponding main text to avoid any perceived over interpretation.

27. What are the niches? How big are they?

We thank the reviewer for the comment. A niche can arise from multiple populations consuming the same substrate. Different substrates would lead to different niches. In the model, different populations in the same niche compete with each other by having a shared carrying capacity. In our simulation, the number of coexisting niches was randomized between 1 and the total number of species. Each species was randomly distributed into one of the niches. The carrying capacities of each niche was randomized following a uniform distribution.

We have updated the main text to clarify the definition and the sizes of the niches. In light of the reviewer’s comment, we also added section 2.1.3 in the supplementary information to describe the details of parameter randomizations in our simulations.

28. Figure 4: Can you add a vertical line for $w = 1$ to aid in the importance of this as a threshold

Thanks for the suggestion. We added the vertical lines for $\omega = 1$ in Fig. 4a and b.

29. Methods: Where are the methods for the model construction / analysis / code?

We thank the reviewer for raising this point. In the updated manuscript, we added the model construction, analysis and simulation in Methods. The Matlab codes are available at https://github.com/youlab/PlasmidPersistencePotential_TengWang.

30. Supplementary Information:

Equations 23 and 24: you mention a q parameter just below these equations, but that term is not in the Equations. I suggest you double-check all equations in case there are other minor typos hidden there.

Thanks for pointing out this typo. We double-checked the main text as well as the supplementary information and corrected the minor typos.

Reviewer #2

Overview:

In the paper “The Persistence Potential of Mobile Genetic Elements” the authors propose a theoretical framework to evaluate the stability of plasmids in microbial communities. As the authors argue, this is an important problem that has been studied since the seminal studies by Levin and Stewart in the 1970s. In this paper, the authors argue that the subpopulation-centered framework used previously to study the population dynamics of plasmid-bearing populations has many limitations and thus cannot be used to study complex microbial communities. This statement is correct, and, for this reason, several generations of mathematical modelers have restrained from using ODE models to study microbial communities composed of multiple strains and plasmids. To circumvent this problem, the authors propose to use a so-called plasmid-centric framework, a modeling approach that describes explicitly the relative abundances of MGEs in the community from the average fitness effects to the host

We appreciate the reviewer for recognizing the importance of the problem and the potential value of our work. We are particularly glad to see that the reviewer shares our view on the limitation of the subpopulation-centered framework, particularly for plasmid dynamics. We also appreciate the reviewer’s critiques on the complexity of our theory and experiments. In light of these critiques, we recognized the need to better clarify the central premise of our analysis:

Our work addressed **two fundamental challenges** facing the quantitative analysis of the gene flow dynamics in microbial communities. First, it has been computationally prohibitive to construct or simulate ODE-based models for complex communities where more than dozen plasmids are transferred. This challenge is resolved by using our plasmid-central modeling framework through drastic dimension reduction in model formulation. Second, the PCF enabled the heuristic derivation of a generic metric that predicts the persistence and prevalence of any transferable plasmid in a microbial community, based on its kinetic parameters and the community composition.

The reviewer raised two important critiques: (1) Even with the significant simplification of our framework, the complete parameterization of complex communities remains impossible; (2) The complexity of synthetic communities in our experiments is not comparable to natural microbiomes. We have revised the manuscript to address these issues, as detailed in the following point-to-point responses.

Issues:

1. This is a potentially interesting approach that reduces the number of equations and parameters needed to model diverse communities with multiple plasmids transferring vertically and horizontally. For instance, the authors make a back-of-the-envelope calculation that, in order to model the microbiome in a bottle of water, it would be necessary to use 10^{56} equations and 10^{114} parameters using the standard modeling framework, while using their approach it would *only* be necessary to use 28,640 equations and “a few kinetic parameters”: 5.2×10^6 parameters values. I strongly agree with the authors that using ODEs to model complex microbial communities is a futile endeavor, but I would argue that this statement also holds for their proposed framework. Indeed, solving non-linear ODE systems with millions of unknown and potentially unidentifiable parameters may be possible from a computational perspective nowadays, but evaluating the robustness of the results to parametric and modeling uncertainties would be absolutely impossible

We thank the reviewer for recognizing the value of our approach. In particular, it is clear that the reviewer recognized the scope of the dimension reduction achieved by our approach. We also agree that even with our reduced model, the model complexity is still large for a realistic community. This critique is important when it comes to **precisely predicting the temporal dynamics of a specific**

community. However, in order to understand the general principle of complex community dynamics, it is often not necessary to quantify all the parameters. Being able to model the system already has intrinsic value even if not many parameters cannot be measured yet. This is indeed a major use of ODE models when describing community dynamics (not addressing HGT). Examples include Coyte KZ et al 2015⁷, Allesina S & Tang S 2012⁸, and Mougi A & Kondoh M 2012⁹, Butler S & O' Dwyer JP 2018¹⁰, Niehaus L et al 2019¹¹ and Angulo MT et al 2019¹². In these examples, the ODE models are also complex and with the majority of parameters not experimentally determined. However, the models were useful for deducing insights.

Using the conventional subpopulation-centric framework, it is computationally infeasible to even simulate the community dynamics, thus hindering the establishment of the general criterion governing the MGE persistence. In our case, the new framework overcomes this challenge and enabled us to derive a general metric to predict the likelihood of persistence for a plasmid. Then, this criterion can be used to guide experimental data interpretation, as illustrated by our results from synthetic communities.

We completely agree with the reviewer that care must be taken when dealing with complex model with a huge number of parameters. However, having the ability to construct and compute such a model can serve as an effective stepping stone toward something that is more intuitive and simpler. For instance, the metric we derived using the new modeling framework (Eq.4) is much more tractable in comparison to the complex models used to test it (Figure 3).

2. To numerically solve their model, the authors make a series of simplifying assumptions (not all stated clearly in the main text), for instance that competition occurs in niches (therefore effectively reducing the size of the community) and that plasmid burden can be estimated from a weighted average of their costs (therefore ignoring pleiotropic effects between plasmids). Moreover, parameters were randomized (although no details are given about the distribution from where parameters are sampled) to perform 2,000 simulations of 5-100 species and 1-50 plasmids for 30,000 hours. The authors argue that both approaches yield similar results

We thank the reviewer for the insightful comment. Indeed, we made a number of assumptions when conducting the simulations. These include:

- Niches: We assumed each community comprises of multiple niches. Different populations in the same niche compete with each other by having a shared carrying capacity. The purpose of assuming niches is to ensure the coexistence of multiple species at steady state. From what we learned from the simulations, the size of the community depends on the species growth rates and the dilution rate. The assumption of niches doesn't reduce the size of the community.
- Plasmid burden: Indeed, when calculating the average plasmid burden, we assumed that there were no complex interactions between plasmids.
- Parameter randomization: In our simulations, the parameters were sampled from uniform distributions.

We have revised our text to further clarify and discuss these assumptions. In light of the reviewer's comment, we also described the details of parameter randomizations in the Methods of the updated main text and the section 2.1.3 of the updated supplementary information.

3. Finally, to evaluate the heuristic expression derived from their model, the authors engineer eight synthetic communities carrying a mobilizable fluorescent plasmid. By performing serial dilution experiments for 15

To put this complexity in perspective, recent studies of synthetic communities have considered no more than 3 plasmids:

- Hall JP et al (2016)¹⁶: In this study, they constructed and modeled a 2-species community transferring **one plasmid**.
- Lopatkin AJ et al (2017)²: The synthetic communities in this study transferred **no more than 3 plasmids**.
- San Millan A et al (2014)¹⁷: The community they modeled transfer only **one plasmid**.
- Zwanzig M et al (2019)¹⁸: The mathematical model in this study described the transfer dynamics of **one single plasmid**.
- Cooper RM et al (2017)¹⁹: In this study, they modeled the transfer of **one single plasmid** in a 2-species community.

The plasmid-centric framework, however, only requires 12 ODEs to model our 5-plasmid community. The formulations of the ODEs are given as follows:

$$\frac{ds_i}{dt} = \alpha_i \mu_i^e s_i - D s_i$$

$$\frac{dp_{i,j}}{dt} = \beta_{i,j} \mu_{i,j}^e p_{i,j} + (s_i - p_{i,j})(\eta_{j,1,i} p_{1,j} + \eta_{j,2,i} p_{2,j}) - (\kappa_{i,j} + D) p_{i,j}$$

where $i = 1,2$ and $j = 1,2,3,4,5$.

That is, even for this relatively simple community (in comparison to natural communities), the reduction in the model complexity by using our new modeling framework is drastic.

In light of the reviewer's comment, we have revised the main text to further clarify the role of the experiments and to put the complexity of our systems in a broader context.

4. In summary, this manuscript addresses an important and significant problem in plasmid biology. However, this is a very difficult paper to follow, as most of the information necessary to understand their model and experimental systems can only be found in the Supplementary Information

We appreciate the reviewer for recognizing the importance and significance of the problem we're trying to address. We regret that our presentation was not sufficiently clear. We have thoroughly revised the text to improve clarity and rigor in our description.

5. Also, introducing new models and biological systems (i.e. phages and transposons) at the end of the manuscript makes it very difficult to understand.

We appreciate the comment, which is also shared by reviewer 1. In light of these suggestions, we have narrowed the scope of our work to focus on plasmids and leave the analysis of other types of MGEs to future work.

6. And while the proposed MGE-centric framework proposed is potentially useful, I believe it is more appropriate for a theoretical journal where the limitations of this modeling approach would be better scrutinized and its benefits better appreciated.

We appreciate the reviewer for realizing that the framework is potentially useful. The limitations and the benefits of our approach were fully addressed in our point-to-point response and the revised manuscript. We note that Nature Communications have published many papers with the same flavor, where quantitative modeling and experiments are integrated to address fundamental biological questions, including microbial community dynamics.

Reviewer #3

Overview

The authors seek to find a solution for the problem of modelling many mobile genetic elements in bacterial populations, and by defining a new framework in which mobile elements and bacterial communities are separated in different ODE's they can reduce the number of ODE's enormously. This is an important addition for theoretical work in which both MGE dynamics and bacterial community dynamics are of interest, such as microbiome of the gut and antimicrobial resistance. The authors derive a persistence potential, and from this they show based on historical data that their modelling framework can determine the fate of MGE's. This work is a continuation on the work of Stewart & Levin, Simonsen et al, and Lopatkin et al. extending it to multi-MGE microbial communities

We thank the reviewer for recognizing the importance of this work and for constructive comments on both the conceptual and technical aspects of our work. We also appreciate the insightful suggestions by the reviewer. We have fully addressed these raised points in the revision.

Issues:

1. The biological definition of the persistence potential is unclear. Especially in line 137-138 it gets confusing as $\omega < 0$ also indicates persistence, but a prevalence below 0 seems unreasonable. Furthermore, the graph Fig 3 C does not go below 0.

We thank the reviewer for the comment. In our modeling framework, $\omega < 0$ arises if plasmid is beneficial and its benefit overcomes the rate of plasmid loss. Under this condition, the plasmid can persist.

Due to the intuitive outcome, we do not expand Figure 3C to include $\omega < 0$. In light of the reviewer's comment, we have revised the main text and the Figure caption to clarify this point.

2. Considering the mathematics it seems to be the value for which a non-negative equilibrium exists. Thus it would have a similar interpretation as the basic reproduction number in ecology and epidemiology. If so it would be a good idea if the authors place the ideas in the context of such widely used concepts. Not necessarily but also of interest might be to consider percolation thresholds such used by Davis et al Nature volume 454, pages634–637(2008)

We thank the reviewer for bringing these key concepts into our attention. Indeed, the concepts of the basic reproduction number and the percolation threshold provide very interesting insights and share similar interpretation as our persistence potential. We have cited the relevant papers and commented on the connection of our work to it.

3. The derivation of the community with diversity lines 146-149 and the supplementary information do not give any argumentation or proof that this correct. If it is a pragmatic choice that worked out properly it is fine, but then please indicate so.

The reviewer is correct that this is a pragmatic choice that worked out properly. Indeed, due to the complexity of the community models in general (even when using the new framework), it is impossible to derive a general analytical solution. However, our approximate metric (by factoring in the community composition) provides an excellent approximation in terms of predicting the plasmid persistence. In light of the reviewer's comment, we have revised the main text to clarify this point.

4. The definition of α μ λ etc needs to be given in the main text. Also the fact that α and λ depend on p_{ij} and s_i must be part of the main text. Otherwise the model cannot be understood from the main text

We thank the reviewer for the suggestion. In the revised manuscript, we have included the definition of these terms in the main text to avoid confusion.

5. Line 35-37: The list of environments is not exhaustive and should be indicated as such. Furthermore addition systems of plasmids are missing.

We thank the reviewer for the suggestion. We have elaborated this statement to be more comprehensive in listing different environments and included description of addition systems.

6. Line 73: Unclear what is exactly meant by "mobilizable" plasmids

We appreciate the reviewer for the comment. In our work, **a mobilizable plasmid is one that can be transferred horizontally via conjugation**. Depending on whether it encodes the transfer machinery, it can be self-transmissible or non self-transmissible. Our modeling framework is applicable for both cases.

We have revised our text to further clarify this point.

7. Line 74: Dimensions of abundance are not given. Later on s_i is given as relative abundance.

We thank the reviewer for pointing it out. Here, "abundance" means the cell density of each population. When we performed the numerical simulations, we normalized the abundances with the combined capacity of all the niches. Thus, the abundance in our simulations is dimensionless. The normalization didn't change the formation of our framework.

We have revised our main text to clarify this point.

8. In Figure 4 B, two estimates with $\omega \approx 10^{-2}$ show a relative abundance above 1, but this is not discussed.

We thank the reviewer for the comment. In figure 4B, there are two points with $\omega \approx 10^{-2}$ show a relative abundance above 1. This is due to the protocols employed to measure plasmid abundance. These two data points were extracted from the previous work (Fischer EA et al. BMC microbiology. 2014). The authors used selective plating to determine the relative abundance of plasmid-carrying cells. In the mixture of the plasmid-carrying cells (denoted as T) and plasmid-free cells (denoted as R), plasmid-free cells were ciprofloxacin-resistant while plasmid-carrying cells were resistant to both ciprofloxacin and cefotaxime. They first measured the total density of cells (R+T) using ciprofloxacin selection. Then, they measured the density of plasmid-carrying cells (T) using double selection (ciprofloxacin and cefotaxime). Ideally, the measured value of T should also be smaller than R+T. However, because (1) the fraction of R was very small; (2) the relative error of selective plating and colony counting was relatively big, the measured value of T sometimes exceeded the measured value of R+T, which is a result of experimental artifact. Therefore, the relative abundances of these two datapoints were higher than 1. We have added the discussion of these two estimates in the supplementary information.

9. Line 420: Why is this method to determine the conjugation efficiency used and not the end point method?

We appreciate the reviewer for the comment. One of the assumptions underlying the end-point method is that the plasmid-carrying cells and plasmid-free cells have the same growth rates²⁰. The fitness cost of the plasmids can introduce a systematic error to the measured conjugation efficiency.

In our experiments, some plasmids, such as R388 and RP4, have relatively large burdens. Therefore, we employed the method established by Lopatkin AJ². This method measured the conjugation efficiency in a short time period with very little cell growth. Therefore, it was more robust to the fitness cost of the plasmids.

References:

- 1 Condit, R., Stewart, F. M. & Levin, B. R. The population biology of bacterial transposons: a priori conditions for maintenance as parasitic DNA. *The American Naturalist* **132**, 129-147 (1988).
- 2 Lopatkin, A. J. *et al.* Persistence and reversal of plasmid-mediated antibiotic resistance. *Nature communications* **8**, 1689 (2017).
- 3 Bergstrom, C. T., Lipsitch, M. & Levin, B. R. Natural selection, infectious transfer and the existence conditions for bacterial plasmids. *Genetics* **155**, 1505-1519 (2000).
- 4 Mc Grath, S. & van Sinderen, D. *Bacteriophage: genetics and molecular biology*. (Horizon Scientific Press, 2007).
- 5 Rakonjac, J., Bennett, N. J., Spagnuolo, J., Gagic, D. & Russel, M. Filamentous bacteriophage: biology, phage display and nanotechnology applications. *Current issues in molecular biology* **13**, 51 (2011).
- 6 Hilbert, M. & López, P. The world's technological capacity to store, communicate, and compute information. *science* **332**, 60-65 (2011).
- 7 Coyte, K. Z., Schluter, J. & Foster, K. R. The ecology of the microbiome: networks, competition, and stability. *Science* **350**, 663-666 (2015).
- 8 Allesina, S. & Tang, S. Stability criteria for complex ecosystems. *Nature* **483**, 205-208 (2012).
- 9 Mougi, A. & Kondoh, M. Diversity of interaction types and ecological community stability. *Science* **337**, 349-351 (2012).
- 10 Butler, S. & O'Dwyer, J. P. Stability criteria for complex microbial communities. *Nature communications* **9**, 1-10 (2018).
- 11 Niehaus, L. *et al.* Microbial coexistence through chemical-mediated interactions. *Nature communications* **10**, 1-12 (2019).
- 12 Angulo, M. T., Moog, C. H. & Liu, Y.-Y. A theoretical framework for controlling complex microbial communities. *Nature communications* **10**, 1-12 (2019).
- 13 Abreu, C. I., Friedman, J., Woltz, V. L. A. & Gore, J. Mortality causes universal changes in microbial community composition. *Nature communications* **10**, 1-9 (2019).
- 14 Friedman, J., Higgins, L. M. & Gore, J. Community structure follows simple assembly rules in microbial microcosms. *Nature ecology & evolution* **1**, 1-7 (2017).
- 15 Gokhale, S., Conwill, A., Ranjan, T. & Gore, J. Migration alters oscillatory dynamics and promotes survival in connected bacterial populations. *Nature communications* **9**, 1-10 (2018).
- 16 Hall, J. P., Wood, A. J., Harrison, E. & Brockhurst, M. A. Source–sink plasmid transfer dynamics maintain gene mobility in soil bacterial communities. *Proceedings of the National Academy of Sciences* **113**, 8260-8265 (2016).
- 17 San Millan, A. *et al.* Positive selection and compensatory adaptation interact to stabilize non-transmissible plasmids. *Nature communications* **5**, 1-11 (2014).
- 18 Zwanzig, M. *et al.* Mobile compensatory mutations promote plasmid survival. *Msystems* **4** (2019).
- 19 Cooper, R. M., Tsimring, L. & Hasty, J. Inter-species population dynamics enhance microbial horizontal gene transfer and spread of antibiotic resistance. *Elife* **6**, e25950 (2017).
- 20 Simonsen, L., Gordon, D., Stewart, F. & Levin, B. R. Estimating the rate of plasmid transfer: an endpoint method. *Microbiology* **136**, 2319-2325 (1990).

REVIEWERS' COMMENTS

Reviewer #1 (Remarks to the Author):

A comprehensive response to my comments, thank you.

Reviewer #2 (Remarks to the Author):

As the authors discuss in the introduction, modeling plasmid dynamics has been an active endeavor since the 1970s. The goal of previous theoretical studies has been to understand the molecular, genetic, or eco-evolutionary mechanisms that enable plasmids to persist in bacterial populations. For this reason, previous studies have been performed in controlled experimental systems and analyzed using simple compartmental models where the fitness effects associated with plasmid carriage and their interaction with the environment can be evaluated explicitly (an approach the authors refer to as a subpopulation-centric framework). In contrast, the goal of this manuscript is to determine a priori if a plasmid would persist in a complex community composed of hundreds of plasmids and hundreds of strains. This is an important and difficult problem, and one that, as argued by the authors, cannot be addressed using the sub-population-centric approach. I agree with the authors, although I also hold that it also cannot be addressed using the plasmid-centric framework described in this manuscript.

The mathematical and computational tools to determine the persistence potential of a plasmid in complex communities are yet to be developed, but it will certainly not result from scaling-up systems of ordinary differential equations to consider thousands of equations with millions of parameters. The reason previous studies have restricted to study simple communities is not because of lack of interest or computational power, but because, independently of the modeling framework used, it's unrealistic to estimate kinetic parameter values for individual strains in complex communities (e.g. the microbiota). The authors claim that estimating 8×10^6 parameters is "both feasible to construct and calculate", which is simply not true. As argued in my previous report, reducing the number of parameters to "only" millions doesn't make a difference when these are impossible to determine empirically (bacteria are not culturable, metabolic interactions are complex and distributions of fitness effects are heterogeneous) or to analyze theoretically (equations are highly non-linear and parameters are not identifiable).

Although not explicitly, the authors agree with this statement, as their model validation is performed using toy model systems composed of a handful of strains and plasmids. In this context, the benefit of their approach is not so clear, as previous models already predicted efficiently and with quantitative accuracy the probability of plasmid maintenance in these communities. Moreover, the plasmid persistence potential expression proposed by the authors provides a similar result as the relative fitness approach used by previous studies (it can be seen in Figure 4 that $w > 1$ roughly corresponds to relative abundance values larger than 0.5, indicating that the plasmid would be under positive selection). Indeed, the expression proposed by the authors showing that plasmids persist when the benefit of bearing plasmids overcomes plasmid loss is analogous to the criterium proposed in the seminal study of Stewart and Levin (not cited in this context).

In summary, although the resubmitted version of the manuscript has improved in clarity and was considerably toned down, I still believe the modeling approach discussed in this manuscript does not apply to realistic scenarios and doesn't provide any insight about the biology of plasmids in microbial communities, therefore representing only a technical contribution to the field and of interest to a narrow community of mathematical modelers.

Reviewer #3 (Remarks to the Author):

I would like to express my apologies, that due to other work and holidays my review has been delayed so much.

Especially because I found that the authors have sufficiently answered all my comments and that I do not have any further remarks.

I recommend that this paper is published.

With kind regards

Egil Fischer

Reviewer #1 (Remarks to the Author):

A comprehensive response to my comments, thank you.

We are extremely grateful that the reviewer believes our updated manuscript has addressed all previous concerns raised.

Reviewer #2 (Remarks to the Author):

As the authors discuss in the introduction, modeling plasmid dynamics has been an active endeavor since the 1970s. The goal of previous theoretical studies has been to understand the molecular, genetic, or evolutionary mechanisms that enable plasmids to persist in bacterial populations. For this reason, previous studies have been performed in controlled experimental systems and analyzed using simple compartmental models where the fitness effects associated with plasmid carriage and their interaction with the environment can be evaluated explicitly (an approach the authors refer to as a subpopulation-centric framework). In contrast, the goal of this manuscript is to determine a priori if a plasmid would persist in a complex community composed of hundreds of plasmids and hundreds of strains. This is an important and difficult problem, and one that, as argued by the authors, cannot be addressed using the sub-population-centric approach. I agree with the authors, although I also hold that it also cannot be addressed using the plasmid-centric framework described in this manuscript.

We appreciate the reviewer for agreeing with the importance and challenge of understanding plasmid persistence potential in complex microbial communities. We also appreciate the author for sharing our view that this problem cannot be addressed using the conventional approach. Indeed, the central premise of this work is to overcome the computational challenge associated with gene flow in complex communities and to develop a general metric to predict plasmid persistence and abundance in microbiota.

The mathematical and computational tools to determine the persistence potential of a plasmid in complex communities are yet to be developed, but it will certainly not result from scaling-up systems of ordinary differential equations to consider thousands of equations with millions of parameters. The reason previous studies have restricted to study simple communities is not because of lack of interest or computational power, but because, independently of the modeling framework used, it's unrealistic to estimate kinetic parameter values for individual strains in complex communities (e.g. the microbiota). The authors claim that estimating 8×10^6 parameters is "both feasible to construct and calculate", which is simply not true. As argued in my previous report, reducing the number of parameters to "only" millions doesn't make a difference when these are impossible to determine empirically (bacteria are not culturable, metabolic interactions are complex and distributions of fitness effects are heterogeneous) or to analyze theoretically (equations are highly non-linear and parameters are not identifiable). Although not explicitly, the authors agree with this statement, as their model validation is performed using toy model systems composed of a handful of strains and plasmids.

We appreciate the reviewer for agreeing that determining the persistence potential of a plasmid in complex communities represents a fundamental challenge. The reviewer commented that the previous studies were restricted primarily by the technical difficulties of estimating kinetic parameters, rather than the lack of computational power. Based on the comment, it is clear that this reviewer recognizes the fundamental challenge associated with modeling a microbial community transferring multiple plasmids using a sub-population-centric model. This is the very challenge that our work has addressed.

We agree that estimating kinetic parameters, especially in complex communities, is valuable and remains a challenge for complex biological systems, including microbial communities. However, the reviewer seems to imply that modeling would only be useful if one could empirically determine all

parameters directly. We respectfully disagree with this notion. Also, we did not claim that estimating 8×10^6 parameters would be currently feasible. Instead, we made two claims:

- 1) Our framework enables construction and calculation of a model for an HGT-mediated microbial community of realistic size.
- 2) Even if many of the parameters cannot be determined experimentally, the ability to carry out such modeling is useful. As we stated in our previous response:

“... However, in order to understand the general principle of complex community dynamics, it is often not necessary to quantify all the parameters. Being able to model the system already has intrinsic value even if not many parameters cannot be measured yet. This is indeed a major use of ODE models when describing community dynamics (not addressing HGT). Examples include Coyte KZ et al 2015¹, Allesina S & Tang S 2012², and Mougi A & Kondoh M 2012³, Butler S & O’Dwyer JP 2018⁴, Niehaus L et al 2019⁵ and Angulo MT et al 2019⁶. In these examples, the ODE models are also complex and with the majority of parameters not experimentally determined. However, the models were useful for deducing insights.”

Both points are supported by our results presented in the manuscript. In particular, pertinent to one point the reviewer made, **we were indeed able to analyze the models theoretically to derive a general metric**. The numerical test of the metric (Figure 3) for complex communities was only feasible by using our new modeling framework.

In this context, the benefit of their approach is not so clear, as previous models already predicted efficiently and with quantitative accuracy the probability of plasmid maintenance in these communities. Moreover, the plasmid persistence potential expression proposed by the authors provides a similar result as the relative fitness approach used by previous studies (it can be seen in Figure 4 that $w > 1$ roughly corresponds to relative abundance values larger than 0.5, indicating that the plasmid would be under positive selection). Indeed, the expression proposed by the authors showing that plasmids persist when the benefit of bearing plasmids overcomes plasmid loss is analogous to the criterion proposed in the seminal study of Stewart and Levin (not cited in this context).

Our work is in part inspired and motivated by pioneering theoretic studies in this field, including Stewart and Levin⁷, which we have cited in our manuscript.

However, our modeling and experimental test of the derived persistence potential represents a major technical and conceptual advance in several aspects.

- (1) Previously, a theoretic criterion for the persistence of a plasmid has only been derived for the system consisting of one species transferring one plasmid (Stewart and Levin 1977)⁷. In contrast, our generic criterion is applicable for a community consisting of multiple species transferring multiple plasmids. In terms of **intuitive interpretation**, our criterion is analogous to that by Stewart and Levin: a plasmid persists if the transfer is sufficiently fast to overcome its burden and loss due to segregation error. However, the form of our criterion is not at all evident from looking at that derived from one-species-one-plasmid system. Yet, this specific form is critical for guiding experimental design and data interpretation, as illustrated from our experimental analysis.

Derivation and simulation-mediated validation of this criterion are enabled by the new modeling framework we have developed, underscoring the value of the ability to construct/analyze a model even if not all parameters can be empirically determined.

- (2) In addition to predict whether a plasmid is likely to persist, we also established how our metric can provide approximate, quantitative prediction on the relative abundance of the plasmids. The latter aspect has not been done previously, even for one species transferring one plasmid.
- (3) In addition to theoretic derivation, we have provided substantial experimental tests of the criterion using both published data and newly generated data from synthetic communities.

In light of the reviewer's comment, we have revised the Discussion section to further clarify these points.

In summary, although the resubmitted version of the manuscript has improved in clarity and was considerably toned down, I still believe the modeling approach discussed in this manuscript does not apply to realistic scenarios and doesn't provide any insight about the biology of plasmids in microbial communities, therefore representing only a technical contribution to the field and of interest to a narrow community of mathematical modelers.

We appreciate the reviewer for realizing that the resubmitted manuscript has been improved. The limitations and the benefits of our approach were fully addressed in our point-to-point responses and the revised manuscript. We note that Nature Communications have published many papers with the same flavor, where quantitative modeling and experiments are integrated to address fundamental biological questions, including microbial community dynamics.

Reviewer #3 (Remarks to the Author):

I would like to express my apologies, that due to other work and holidays my review has been delayed so much. Especially because I found that the authors have sufficiently answered all my comments and that I do not have any further remarks. I recommend that this paper is published.

We appreciate the reviewer for believing that our updated manuscript has sufficiently answered all the previous concerns and we are grateful for the recommendation for publication.

References

- 1 Coyte, K. Z., Schluter, J. & Foster, K. R. The ecology of the microbiome: networks, competition, and stability. *Science* **350**, 663-666 (2015).
- 2 Allesina, S. & Tang, S. Stability criteria for complex ecosystems. *Nature* **483**, 205-208 (2012).
- 3 Mougi, A. & Kondoh, M. Diversity of interaction types and ecological community stability. *Science* **337**, 349-351 (2012).
- 4 Butler, S. & O'Dwyer, J. P. Stability criteria for complex microbial communities. *Nature communications* **9**, 1-10 (2018).
- 5 Niehaus, L. *et al.* Microbial coexistence through chemical-mediated interactions. *Nature communications* **10**, 1-12 (2019).
- 6 Angulo, M. T., Moog, C. H. & Liu, Y.-Y. A theoretical framework for controlling complex microbial communities. *Nature communications* **10**, 1-12 (2019).
- 7 Stewart, F. M. & Levin, B. R. The population biology of bacterial plasmids: a priori conditions for the existence of conjugationally transmitted factors. *Genetics* **87**, 209-228 (1977).